# KDM6B interacts with TFDP1 to activate P53 signaling in regulating mouse palatogenesis

Tingwei Guo, Xia Han, Jinzhi He, Jifan Feng, Junjun Jing, Eva Janečková, Jie Lei, Thach-Vu Ho, Jian Xu, Yang Chai*

Center for Craniofacial Molecular Biology, Herman Ostrow School of Dentistry, University of Southern California, Los Angeles, United States

**Abstract** Epigenetic regulation plays extensive roles in diseases and development. Disruption of epigenetic regulation not only increases the risk of cancer, but can also cause various developmental defects. However, the question of how epigenetic changes lead to tissue-specific responses during neural crest fate determination and differentiation remains understudied. Using palatogenesis as a model, we reveal the functional significance of *Kdm6b*, an H3K27me3 demethylase, in regulating mouse embryonic development. Our study shows that *Kdm6b* plays an essential role in cranial neural crest development, and loss of *Kdm6b* disturbs P53 pathway-mediated activity, leading to complete cleft palate along with cell proliferation and differentiation defects in mice. Furthermore, activity of H3K27me3 on the promoter of *Trp53* is antagonistically controlled by *Kdm6b*, and *Ezh2* in cranial neural crest cells. More importantly, without *Kdm6b*, the transcription factor TFDP1, which normally binds to the promoter of *Trp53*, cannot activate *Trp53* expression in palatal mesenchymal cells. Furthermore, the function of *Kdm6b* in activating *Trp53* in these cells cannot be compensated for by the closely related histone demethylase *Kdm6a*. Collectively, our results highlight the important role of the epigenetic regulator KDM6B and how it specifically interacts with TFDP1 to achieve its functional specificity in regulating *Trp53* expression, and further provide mechanistic insights into the epigenetic regulatory network during organogenesis.

*For correspondence:
ychai@usc.edu

## Editor's evaluation

Using the mouse secondary palate as a model, this study reports original findings on the function of Kdm6b, a H3K27me3 demethylase, in the regulation of embryonic development. The authors show that Kdm6b plays an essential role in neural crest development, and that loss of Kdm6b perturbs the p53 pathway, leading to complete clefting of the secondary palate along with cell proliferation and differentiation defects.

## Introduction

Embryonic development is a highly complex self-assembly process during which precursor cells are coordinated to generate appropriate cell types and assemble them into well-defined structures, tissues, and organs (*Shahbazi et al., 2016*). During this process, precursor cells undergo extensive and rapid cell proliferation until they reach the point of exit from the cell cycle to differentiate into various cell lineages (*Ruijtenberg and van den Heuvel, 2016*; *Miermont et al., 2019*). How these precursor cells modulate expression of different genes and proceed through diverse proliferation and differentiation processes is a very complex and interesting question. Growing evidence shows that epigenetic regulation, which includes mechanisms such as DNA methylation, histone modifications,

chromatin accessibility, and higher-order organization of chromatin, provides the ability to modify gene expression and associated protein production in a cell type-specific manner, thus playing an essential role in achieving signaling specificity and regulating cell fate during embryonic development (*Hanna et al., 2018*).

Among these various layers of epigenetic regulation, DNA methylation and histone methylation are the best characterized and known to be key regulators of diverse cellular events (*Bannister and Kouzarides, 2011*; *Smith and Meissner, 2013*; *Molina-Serrano et al., 2019*). For example, methylation of lysine 27 on histone H3 (H3K27me) by methyltransferases is a feature of heterochromatin that renders it inaccessible to transcription factors, thus maintaining transcriptional repression, across many species (*Wiles and Selker, 2017*). On the other hand, methylation of H3K4me3 found near the promoter region can couple with the NURF complex to increase chromatin accessibility for gene activation (*Wysocka et al., 2006*; *Soares et al., 2017*). Demethylation, which results from removing a methyl group, also plays important roles during development. For instance, demethylation of H3K4 is required for maintaining pluripotency in embryonic stem cells, and demethylases KDM6A and KDM6B are required for proper gene expression in mature T cells (*Lessard and Crabtree, 2010*; *Jambhekar et al., 2019*). These studies clearly show that failure to maintain epigenomic integrity can cause deleterious consequences for embryonic development and adult tissue homeostasis (*Henckel et al., 2007*; *Kim et al., 2009*; *Kang et al., 2019*).

Palatogenesis is a complex process known to be regulated by multiple genetic regulatory mechanisms, including several signaling pathways (BMP, SHH, WNT, FGF, and TGFβ) and different transcription factors (such as *Msx1*, *Sox9*, *Lhx6/8*, *Dlx5*, and *Shox2*) (*Satokata and Maas, 1994*; *Yu et al., 2005*; *Chai and Maxson, 2006*; *Levi et al., 2006*; *Cobourne et al., 2009*; *Lee and Saint-Jeannet, 2011*; *Nakamura et al., 2011*; *Bush and Jiang, 2012*; *He and Chen, 2012*; *Parada and Chai, 2012*; *Xu et al., 2016*; *Reynolds et al., 2019*). However, environmental effects can also contribute to orofacial defects, which lends further support to the notion that genetic factors are not sufficient to fully explain the etiology of many birth defects (*Dixon et al., 2011*; *Roessler et al., 2012*; *Seelan et al., 2012*). Furthermore, case studies have revealed that heterozygous mutation of a chromatin-remodeling factor, *SATB2*, and variation in DNA methylation can cause cleft palate in patients (*Leoyklang et al., 2007*; *Chandrasekharan and Ramanathan, 2014*; *Young et al., 2021*). These cases have drawn our attention to the function of epigenetic regulation in palatogenesis.

The contribution of cranial neural crest cells (CNCCs) is critical to palate mesenchyme formation. Recently, studies have begun to address the role of epigenetic regulation in neural crest cell fate determination during development. For instance, homozygous loss of *Arid1a*, a subunit of SWI/SNF chromatin remodeling complex, in neural crest cells results in lethality in mice, associated with severe defects in the heart and craniofacial bones (*Chandler and Magnuson, 2016*). In addition, both lysine methyltransferase *Kmt2a* and demethylase *Kdm6a* are essential for cardiac and neural crest development (*Shpargel et al., 2017*; *Sen et al., 2020*). However, how these epigenetic changes lead to tissue-specific response during neural crest fate determination remain to be elucidated.

In this study, using palatogenesis as a model we investigated the functional significance of the demethylase *Kdm6b* in regulating the fate of CNCCs during palatogenesis. We have discovered that loss of *Kdm6b* in cranial neural crest (CNC)-derived cells results in complete cleft palate along with soft palate muscle defects. We also found cell proliferation and differentiation defects of CNC-derived cells in *Kdm6b* mutant mice. More importantly, our study shows that the level of H3K27me3 on the promoter of *Trp53* (also known as P53 in human) is antagonistically controlled by *Kdm6b* and *Ezh2*. Furthermore, without *Kdm6b*, the transcription factor TFDP1, which binds to the promoter of *Trp53*, cannot activate expression of *Trp53* in palatal mesenchymal cells. More importantly, the function of *Kdm6b* in activating *Trp53* in these cells cannot be compensated for by the closely related histone demethylase *Kdm6a*. Our study highlights the importance of epigenetic regulation on cell fate decision and its function in regulating activity of *Trp53* in CNC-derived cells during organogenesis.

## Results

### Loss of *Kdm6b* in CNC-derived cells results in craniofacial malformations

Previous research has shown that the X-chromosome-linked H3K27 demethylase KDM6A is indispensable for neural crest cell differentiation and viability as it establishes appropriate chromatin structure (*Schwarz et al., 2014*; *Shpargel et al., 2017*). However, we do not yet have a comprehensive understanding of the roles of two other members of the KDM6 family, *Kdm6b* and *Uty*, in regulating CNCCs during craniofacial development. More importantly, we have yet to understand how demethylase achieves its functional specificity in regulating downstream target genes. In order to elucidate the functions of *Kdm6b* and *Uty*, we first evaluated the expression patterns of KDM6 family members in the palatal region (*Figure 1—figure supplement 1A–F*). We found that, of these, *Kdm6b* is more abundantly expressed than *Kdm6a* and *Uty* in both palate mesenchymal and epithelial cells, which indicated it might play a critical role in regulating palatogenesis.

To investigate the tissue-specific function of *Kdm6b* during craniofacial development, we generated *Wnt1^Cre^;Kdm6b^fl/fl^* and *Krt14^Cre^;Kdm6b^fl/fl^* mice to specifically target the deletion of *Kdm6b* in CNC-derived and epithelial cells, respectively. Loss of *Kdm6b* in CNC-derived cells resulted in complete cleft palate in *Wnt1^Cre^;Kdm6b^fl/fl^* mice (90% phenotype penetrance, N = 10) and postnatal lethality at newborn stage (100% phenotype penetrance, N = 10) without interrupting expression of other KDM6 family members ( *Figure 1A and B*, *Figure 1—figure supplement 1A–N*). To evaluate when *Kdm6b* was inactivated in the CNC-derived cells, we also investigated the expression of *Kdm6b* at E9.5, well prior to the formation of the palate primordium, and found that *Kdm6b* was efficiently inactivated in the CNC-derived cells at this stage (*Figure 1—figure supplement 1O and P*). Interestingly, loss of *Kdm6b* in epithelial cells did not lead to obvious defects in the craniofacial region in *Krt14^Cre^;Kdm6b^fl/fl^* mice (*Figure 1—figure supplement 2A–H*). These results emphasized that *Kdm6b* is specifically required in CNC-derived cells during palatogenesis. CT images also confirmed the complete cleft palate phenotype and revealed that the most severe defects in the palatal region of *Wnt1^Cre^;Kdm6b^fl/fl^* mice were hypoplastic palatine processes of the maxilla and palatine bones (*Figure 1C and D*). Except for a minor flattened skull, other CNC-derived bones did not show significant differences between control and *Wnt1^Cre^;Kdm6b^fl/fl^* mice (*Figure 1—figure supplement 2I–N*). To evaluate the phenotype in more detail, we performed histological analysis and found that although the palatal shelves were able to elevate, the maxilla and palatine bones, as well as the palate stromal mesenchyme and soft palate muscles, failed to grow towards the midline in *Wnt1^Cre^;Kdm6b^fl/fl^* mice (*Figure 1G–R*, *Figure 1—figure supplement 3A–P*). Furthermore, in the posterior soft palate region, *Wnt1^Cre^;Kdm6b^fl/fl^* mice also showed morphological defects related to the orientation of muscle fibers and the pterygoid plate (*Figure 1—figure supplement 3A–X*). However, since the soft palate forms subsequent to the hard palate, it is difficult to identify whether the soft palatal muscle phenotype is a primary defect or a consequence resulting from an anterior cleft. Therefore, we focused on the anterior hard palate for further investigation. Collectively, these data indicate that mesenchymal *Kdm6b* is indispensable for craniofacial development and plays an essential role during palatogenesis.

### *Kdm6b* is critical for proliferation and differentiation of CNC-derived palatal mesenchymal cells

During craniofacial development, CNCCs migrate ventrolaterally and populate the branchial arches to give rise to distinct mesenchymal structures in the head and neck, such as the palate. Failure of CNCCs to populate pharyngeal arches causes craniofacial defects (*Noden, 1983*; *Noden, 1991*; *Trainor and Krumlauf, 2000*; *Cordero et al., 2011*). To determine whether *Kdm6b* mutant CNCCs successfully populate the first pharyngeal arch, which gives rise to the palatal shelves, we generated tdTomato reporter mice and collected samples at E10.5. The results showed that CNCCs' migration was not adversely affected in *Kdm6b* mutant mice (*Figure 2—figure supplement 1A and B*). Then, we evaluated the process of palatogenesis at different embryonic stages and found that the cleft palate phenotype emerged as early as E14.5 in *Wnt1^Cre^;Kdm6b^fl/fl^* mice (*Figure 2—figure supplement 1C and D*). These data established that *Kdm6b* is not essential for CNCCs entering the pharyngeal arch but is specifically required in regulating post-migratory CNC-derived cells during palatogenesis.

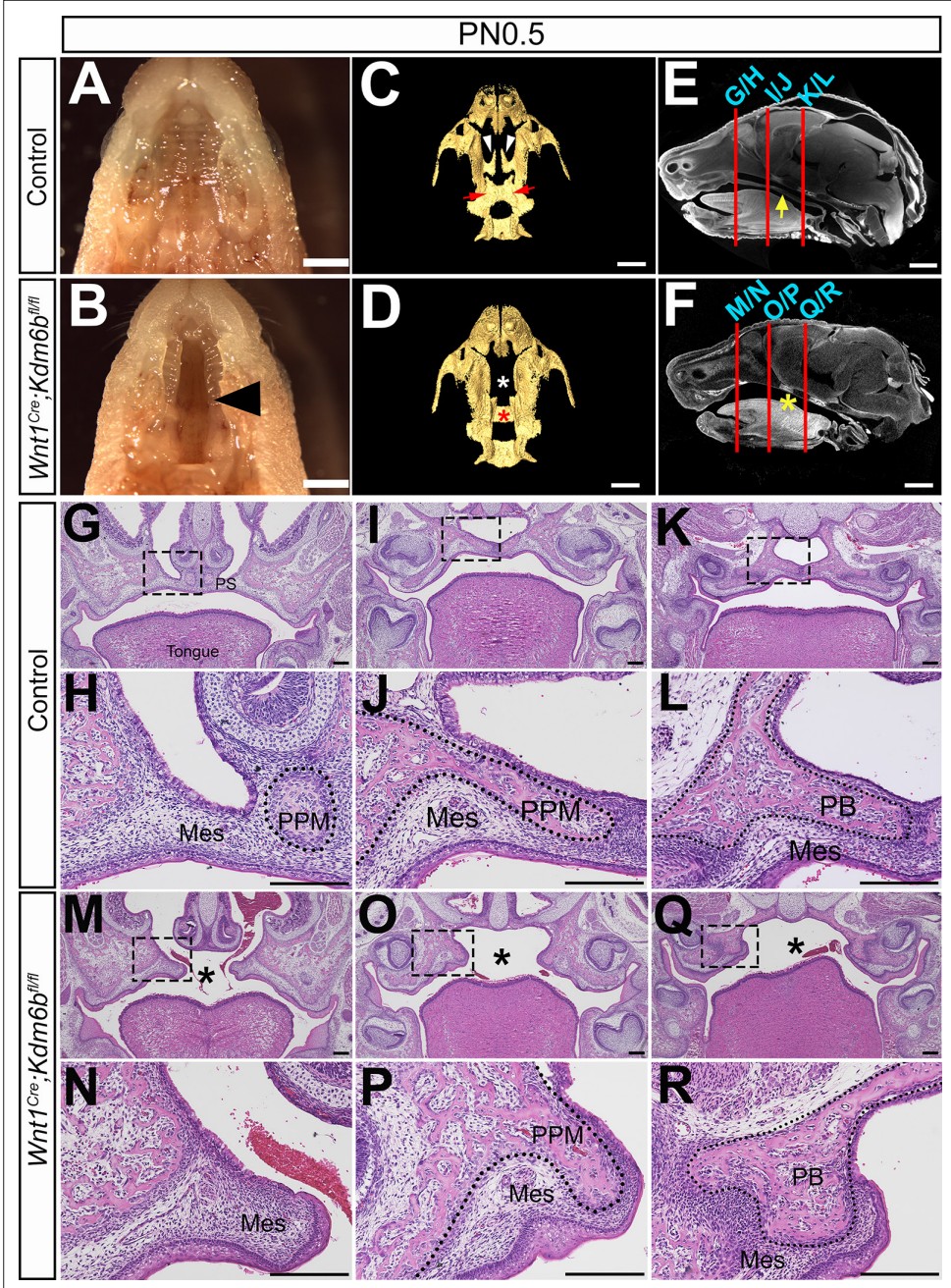

**Figure 1.** Loss of *Kdm6b* in cranial neural crest (CNC)-derived cells results in cleft palate. (**A, B**) Whole-mount oral view shows complete cleft palate phenotype in *Wnt1^Cre^;Kdm6b^fl/fl^* mice. Arrowhead in (**B**) indicates the cleft palate. Scale bar: 2 mm. (**C, D**) CT imaging reveals that the palatine process of the maxilla and palatine bone (PB) is missing in *Wnt1^Cre^;Kdm6b^fl/fl^* mice. White arrowheads in (**C**) indicate the palatine process of maxilla (PPM) in control mice, and red arrows indicate the PB. White asterisk in (**D**) indicates the missing PPM in *Wnt1^Cre^;Kdm6b^fl/fl^* mice, and red asterisk in (**D**) indicates the missing PB in *Kdm6b* mutant mice. Scale bars: 1 mm. (**E, F**) Sagittal views of CT images demonstrate the locations of HE sections in (**G–R**). Red lines indicate the locations of sections. Yellow arrow in (**E**) indicates palatal shelf, and yellow asterisk in (**F**) indicates cleft. Scale bars: 1 mm. (**G–R**) Histological analysis of control and *Wnt1^Cre^;Kdm6b^fl/fl^* mice. (**H, J, L, N, P, R**) are magnified images of boxes in (**G, I, K, M, O, Q**), respectively. Asterisks in (**M, O, Q**) indicate cleft in *Kdm6b* mutant mice. Scale bar: 200 µm. Mes: mesenchyme.

The online version of this article includes the following source data and figure supplement(s) for figure 1:

**Figure supplement 1.** Expression of KDM6 family.

**Figure supplement 1—source data 1.** Source data for *Figure 1—figure supplement 1M*.

*Figure 1 continued on next page*

**Figure supplement 2.** Loss of *Kdm6b* in epithelium (Epi) and cranial neural crest (CNC)-derived cells.

**Figure supplement 3.** Loss of *Kdm6b* in cranial neural crest (CNC)-derived cells results in soft palate muscle defects.

Because cell proliferation defects in CNC-derived cells frequently lead to craniofacial defects, we tested whether loss of *Kdm6b* can affect cell proliferation using EdU labeling. After 2 hr of EdU labeling, we found that the number of cells positively stained with EdU was significantly increased in the CNC-derived palatal mesenchyme in *Wnt1^{Cre};Kdm6b^{fl/fl}* mice compared to controls (*Figure 2A–C*). In addition, after 48 hr of EdU labeling, we found that the number of Ki67 and EdU double-positive cells was significantly increased in *Wnt1^{Cre};Kdm6b^{fl/fl}* mice (*Figure 2D–H*). These results indicated that loss of *Kdm6b* in CNC-derived cells resulted in more cells remaining in the cell cycle and actively proliferating, which further led to hyperproliferation of mesenchymal cells in the palatal region of *Wnt1^{Cre};Kdm6b^{fl/fl}* mice. Meanwhile, palatal mesenchymal cells of *Wnt1^{Cre};Kdm6b^{fl/fl}* mice showed more expression of β-galactosidase, which suggested increased cellular senescence in *Wnt1^{Cre};Kdm6b^{fl/fl}* mice (*Figure 2—figure supplement 1E–G*). To evaluate cellular senescence in vivo, we stained Lamin B1 in EdU-labeled samples. After 48 hr of EdU labeling, we found that there was less expression of Lamin B1 in the palatal mesenchyme of *Wnt1^{Cre};Kdm6b^{fl/fl}* mice. More importantly, fewer EdU+ cells expressing Lamin B1 were observed in *Wnt1^{Cre};Kdm6b^{fl/fl}* mice (*Figure 2—figure supplement 1H–L*). These data suggested that hyperproliferation may cause increased cellular senescence in palatal mesenchymal cells of *Wnt1^{Cre};Kdm6b^{fl/fl}* mice.

Typically, cell proliferation and differentiation are inversely correlated. Differentiation of precursor cells is generally associated with arrested proliferation and permanently exiting the cell cycle (*Ruijtenberg and van den Heuvel, 2016*). To test whether cell differentiation was affected in the CNC-derived palatal mesenchyme in *Wnt1^{Cre};Kdm6b^{fl/fl}* mice, we examined the distribution of the early osteogenesis marker RUNX2 and the later osteogenesis marker SP7 in the palatal region from E13.5 to E15.5 (*Figure 2I–R*). There was a decrease in the number of RUNX2+ cells in the palatal mesenchyme at both E13.5 and E14.5 in *Wnt1^{Cre};Kdm6b^{fl/fl}* mice in comparison to the control (*Figure 2I–L and Q*). In addition, SP7+ cells were also decreased in *Wnt1^{Cre};Kdm6b^{fl/fl}* mice at both E14.5 and E15.5 (*Figure 2M–P and R*). Furthermore, when we induced osteogenic differentiation in palatal mesenchymal cells from E13.5 embryos for 3 weeks, we found that cells from *Wnt1^{Cre};Kdm6b^{fl/fl}* mice showed much less calcium deposition than cells from control mice, indicating a reduction in osteogenic potential in cells from *Kdm6b* mutant mice (*Figure 2S–W*). These results indicated that *Kdm6b* was indispensable for maintaining normal proliferation and differentiation of CNC-derived cells.

## Loss of *Kdm6b* in CNC-derived cells disturbs P53 pathway-mediated activity

In order to identify the downstream targets of *Kdm6b* in the palatal mesenchyme, we performed RNA-seq analysis of palatal tissue at E12.5. The results showed that more genes were downregulated than upregulated in the palatal mesenchyme in *Wnt1^{Cre};Kdm6b^{fl/fl}* mice (*Figure 3A*), which is consistent with the function of *Kdm6b* in removing the repressive mark H3K27me3. We further used Ingenuity Pathway Analysis (IPA) and Gene Ontology (GO) analysis to analyze the pathways that were most disturbed in the palatal mesenchyme in *Kdm6b* mutant mice. Surprisingly, both analyses indicated that pathways involving *Trp53* might be disturbed in the palatal mesenchyme in *Wnt1^{Cre};Kdm6b^{fl/fl}* mice (*Figure 3B and C*).

The tumor suppressor P53 plays prominent roles in regulating DNA damage response, including arresting cell growth for DNA repair, directing cellular senescence, and activating apoptosis (*Mijit et al., 2020*). Mutation of *Trp53* is a major cause of cancer development (*Williams and Schumacher, 2016*). Previous research has shown that some homozygous *Trp53* mutant mice exhibit craniofacial defects, including cleft palate, while inappropriate activation of *Trp53* during embryogenesis also causes developmental defects, including craniofacial abnormalities (*Tateossian et al., 2015*; *Bowen et al., 2019*). These results suggest that precise dosage of *Trp53* is indispensable for craniofacial development. We analyzed the expression of *Trp53* in our samples and found that it significantly

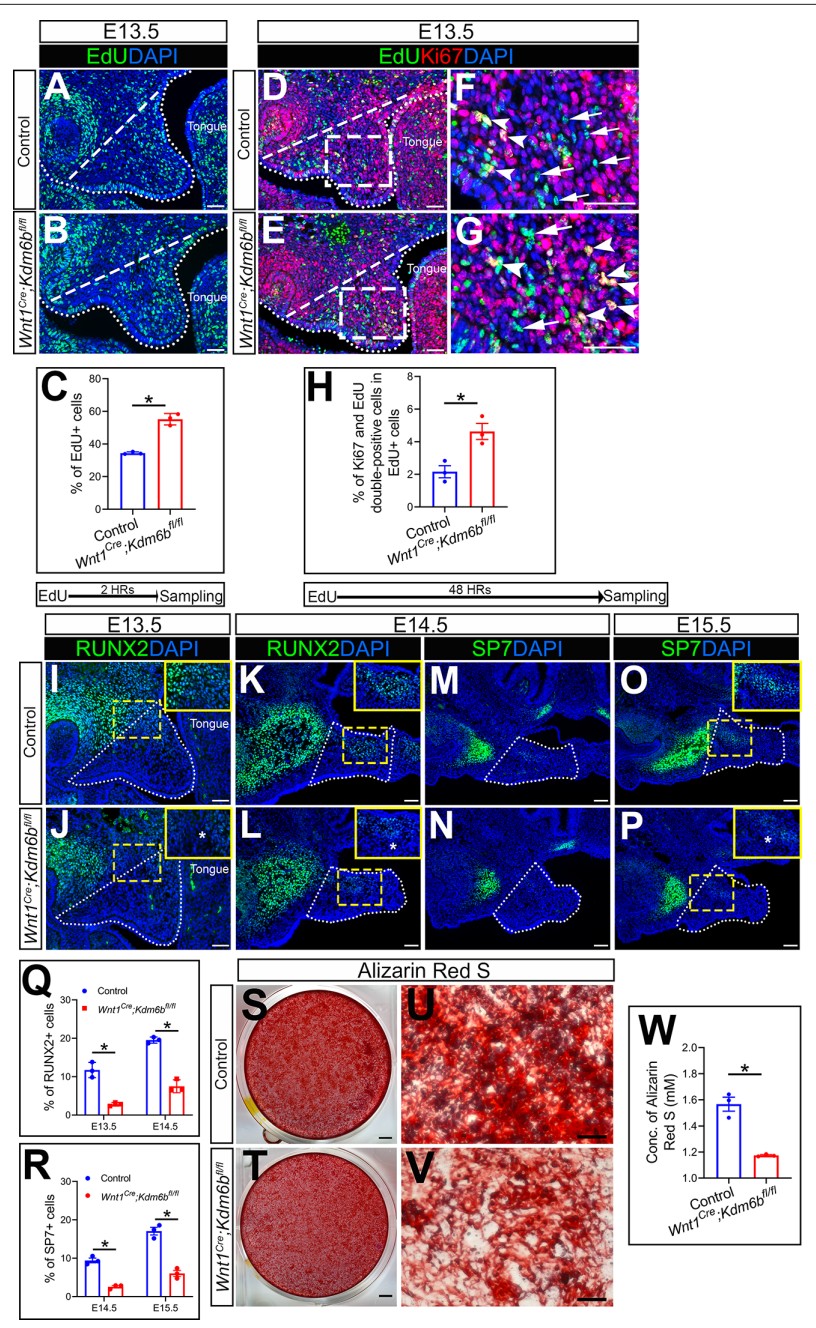

**Figure 2.** *Kdm6b* is critical for proliferation and differentiation of cranial neural crest (CNC)-derived palatal mesenchyme cells. (**A, B**) Immunostaining of EdU at E13.5 after 2 hr of EdU labeling. Dotted lines indicate palatal shelf region. Dashed lines indicate the palatal region used for quantification in (**C**). Scale bar: 50 μm. (**C**) Quantification of EdU+ cells represented in (**A, B**). *p<0.05. (**D–G**) Co-localization of EdU and Ki67 at E13.5 after 48 hr of EdU labeling. Dotted lines indicate palatal shelf region. Dashed lines indicate the palatal region used for quantification in (**H**). (**F**, **G**) are magnified images of boxes in (**D**, **E**). Arrows in (**F**, **G**) indicate representative cells that are only EdU+, while arrowheads indicate representative cells that are positive for both EdU and Ki67. Scale bar: 50 μm. (**H**) Quantification of EdU and Ki67 double-positive cells represented in (**D**, **E**). *p<0.05. (**I–L**) Immunostaining of RUNX2 at indicated stages. Insets are higher-magnification images of boxes in (**I–L**). Asterisks in (**J**, **L**) indicate decreased RUNX2+ cells observed in *Wnt1^{Cre};Kdm6b^{fl/fl}* mice. Scale bar: 50 μm. White dotted lines indicate the palatal region used for quantification in (**Q**). (**M–P**) Immunostaining of SP7 at indicated stages. Insets are higher-magnification images of boxes in (**O**, **P**). Asterisk in (**P**) indicates decreased SP7+ cells observed in *Wnt1^{Cre};Kdm6b^{fl/fl}* mice. Scale bar: 50 μm. White dotted lines indicate the palatal region used for quantification

*Figure 2 continued on next page*

*Figure 2 continued*

in (**R**). (**Q, R**) Quantification results for RUNX2+ and SP7+ cells represented in (**I–P**). *p<0.05. (**S–W**) Osteogenic differentiation assay using Alizarin red S staining. (**W**) is the quantification result of Alizarin red S staining represented in (**S, T**). Scale bars: 2 mm in (**S, T**); 200 μm in (**U, V**). *p<0.05.

The online version of this article includes the following source data and figure supplement(s) for figure 2:

**Source data 1.** Source data for *Figure 2C*.

**Source data 2.** Source data for *Figure 2H*.

**Source data 3.** Source data for *Figure 2Q*.

**Source data 4.** Source data for *Figure 2R*.

**Source data 5.** Source data for *Figure 2W*.

**Figure supplement 1.** *Kdm6b* is not required for cranial neural crest cells (CNCCs) to populate pharyngeal arches but is critical for survival of palatal mesenchymal cells.

**Figure supplement 1—source data 1.** Source data for *Figure 2—figure supplement 1G*.

**Figure supplement 1—source data 2.** Source data for *Figure 2—figure supplement 1L*.

decreased in the palatal region of the *Kdm6b* mutant mice (*Figure 3D–H*). These results indicate that *Kdm6b* plays an important role in regulating the P53 pathway in the CNC-derived mesenchyme during palatogenesis. To further evaluate the consequence of downregulated *Trp53* in *Wnt1^Cre^;Kdm6b^fl/fl^* mice, we assessed DNA damage, which are tightly related to the function of *Trp53*, in our study. We found that DNA damage increased, as indicated by γH2AX expression, in the palatal mesenchyme in *Wnt1^Cre^;Kdm6b^fl/fl^* mice (*Figure 3I–M*). More importantly, we observed significantly increased γH2AX foci in the EdU+ cells of *Wnt1^Cre^;Kdm6b^fl/fl^* palatal mesenchyme (*Figure 3N–R*). These data indicated that actively proliferating cells in *Wnt1^Cre^;Kdm6b^fl/fl^* mice experienced more severe DNA damage compared to those in the control mice, which might be the result of replication stress caused by the hyperproliferation we observed in *Wnt1^Cre^;Kdm6b^fl/fl^* mice.

## Altered *Trp53* expression is responsible for the developmental defects in *Wnt1Cre;Kdm6b*fl/fl mice

To further test whether downregulated expression of *Trp53* is a key factor in the developmental defects we observed in *Wnt1^Cre^;Kdm6b^fl/fl^* mice, we transfected palatal mesenchymal cells from control mice with siRNA to knock down *Trp53*. qPCR revealed that the expression of *Trp53* was significantly decreased in the cells treated with siRNA after 3 days (*Figure 4—figure supplement 1A*). At the same time, the group transfected with siRNA for *Trp53* showed a significant increase in EdU+ cells (*Figure 4A–C*). In addition, significantly increased γH2AX+ cells were also observed in the group transfected with siRNA for *Trp53* (*Figure 4D–F*). These data suggested that downregulated expression of *Trp53* in the palatal mesenchymal cells is a key factor that led to the hyperproliferation and increased DNA damage we observed in *Wnt1^Cre^;Kdm6b^fl/fl^* mice. Furthermore, expression of both *Runx2* and *Sp7* was also significantly reduced in the palatal mesenchymal cells transfected with siRNA for *Trp53* (*Figure 4—figure supplement 1B and C*), which indicated that the downregulated expression of *Trp53* in the palatal mesenchymal cells resulted in differentiation defects, which were also observed in *Wnt1^Cre^;Kdm6b^fl/fl^* mice.

To further investigate the function of *Trp53* in *Wnt1^Cre^;Kdm6b^fl/fl^* mice, we tried to increase P53 in *Kdm6b* mutant mice using available small molecules. Previous research showed that MDM2, a ubiquitin ligase, specifically targets P53 for degradation and there is increased P53 activity in *Mdm2* mutant mice, which exhibit a range of developmental defects (*Arya et al., 2010*; *Bowen and Attardi, 2019*; *Bowen et al., 2019*). Nutlin-3, an MDM2 inhibitor that can specifically interrupt interaction between MDM2 and P53, increases P53 in mouse primary neural stem progenitor cells and rescues neurogenic deficits in *Fmr1* KO mice (*Li et al., 2016*). We treated pregnant mice with Nutlin-3 at a dosage based on their body weight at E10.5, E12.5, and E14.5 of pregnancy and then collected samples at E16.5 for analysis. To assess the potential influence of the solvent used to dissolve Nutlin-3 (10% DMSO in corn oil), we also treated mice with 10% DMSO in corn oil at the same embryonic stages. None of the *Kdm6b* mutant mice were rescued after this treatment (N = 3) (*Figure 4—figure supplement 1D and E*). In contrast, Nutlin-3 treatment successfully rescued the cleft palate in three

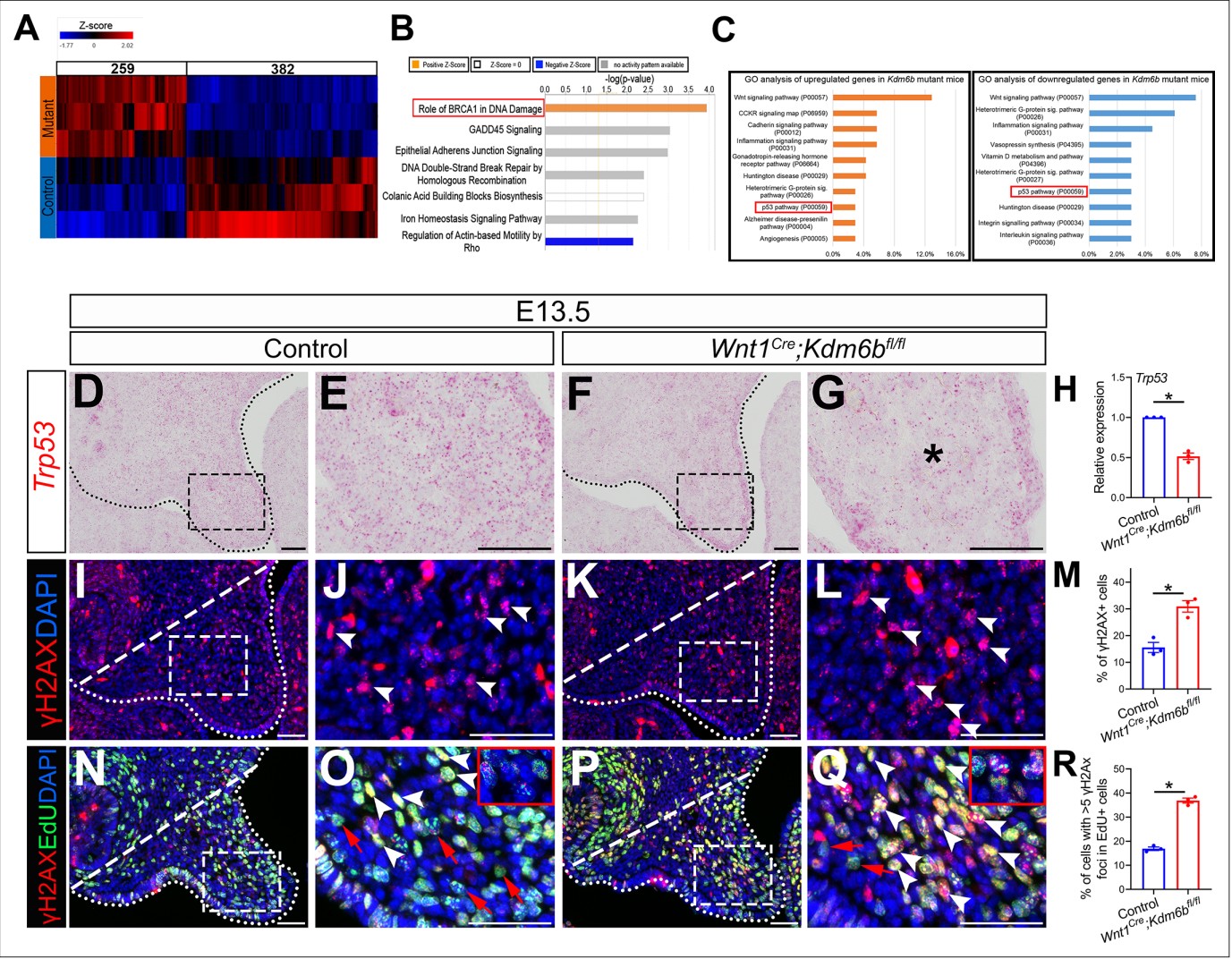

**Figure 3.** P53 signaling pathway is disturbed in *Wnt1^Cre^;Kdm6b^fl/fl^* mice. (**A**) Bulk RNA-seq of palatal tissues collected at E12.5 represented in heatmap. Differentially expressed genes were selected using p<0.05 and fold change <−1.2 or >1.2. (**B**) Top seven signaling pathways disturbed in *Kdm6b* mutant mice, identified by Ingenuity Pathway Analysis. Red box indicates the top upregulated pathway observed in *Wnt1^Cre^;Kdm6b^fl/fl^* sample. (**C**) Top 10 signaling pathways identified by Gene Ontology analysis using differentially expressed genes identified by bulk RNA-seq analysis. Red box indicates P53 signaling is one of the top 10 pathways. X-axis shows the percentage of genes hit against total number of pathways hit. (**D–G**) Expression of *Trp53* at E13.5 using RNAscope in situ hybridization. Dotted lines in (**D, F**) indicate palatal shelf. (**E, G**) are magnified images of boxes in (**D, F**). Asterisk in (**G**) indicates decreased expression of *Trp53* observed in *Wnt1^Cre^;Kdm6b^fl/fl^* mice. Scale bar: 50 μm. (**H**) RT-qPCR quantification of *Trp53* in palatal tissues collected at E13.5. *p<0.05. (**I–L**) Immunostaining of γH2AX at E13.5. Dotted lines in (**I, K**) indicate palatal shelf and dashed lines indicate quantification area. (**J, L**) are magnified images of boxes in (**I, K**), respectively. Arrowheads in (**J, L**) indicate representative γH2AX+ cells. Scale bar: 50 μm. (**M**) Quantification of γH2AX+ cells represented in (**I, K**). *p<0.05. (**N–Q**) Co-localization of EdU and γH2AX at E13.5 after 2 hr of EdU labeling. Dotted lines in (**N, P**) indicate palatal shelf region, while dashed lines indicate the palatal region used for quantification in (**R**). (**O, Q**) are magnified images of boxes in (**N, P**), respectively. Red arrows in (**O, Q**) indicate representative EdU+ cells with less than five γH2AX foci, while white arrowheads indicate representative cells that are positive for EdU and with greater than five γH2AX foci. Scale bar: 50 μm. (**R**) Quantification of EdU+ cells with greater than five γH2AX foci represented in (**N, P**).*p<0.05.

The online version of this article includes the following source data for figure 3:

**Source data 1.** Source data for *Figure 3H*.

**Source data 2.** Source data for *Figure 3M*.

**Source data 3.** Source data for *Figure 3R*.

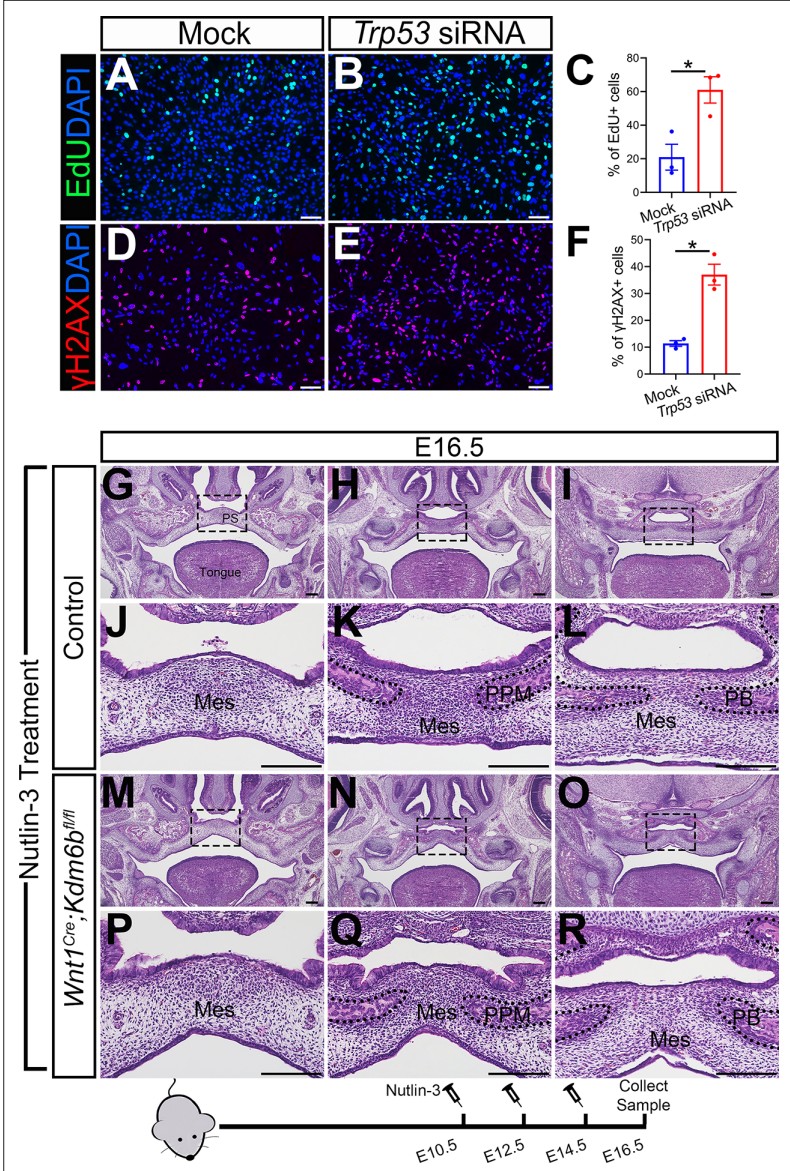

**Figure 4.** Altered *Trp53* expression is responsible for the developmental defects in *Wnt1^Cre^;Kdm6b^fl/fl^* mice. (**A–C**) Cells collected from E13.5 palatal tissue are transfected with siRNA to knock down expression of *Trp53*. Cell proliferation is evaluated using EdU labeling 3 days after transfection. (**A, B**) show proliferation of cells assessed by EdU labeling. Difference in EdU+ cells between mock- and siRNA-transfected groups is quantified in (**C**). Scale bar: 100 µm. *p<0.05. (**D–F**) Cells collected from E13.5 palatal tissue are transfected with siRNA to knock down expression of *Trp53*. DNA damage is evaluated using γH2AX 3 days after transfection. (**D, E**) show γH2AX+ cells. Difference in γH2AX+ cells between mock- and siRNA-transfected groups is quantified in (**F**). Scale bar: 100 µm. *p<0.05. (**G–R**) Histological analysis of control and *Wnt1^Cre^;Kdm6b^fl/fl^* mice treated with Nutlin-3. (**J–L, P–R**) are magnified images of boxes in (**G–I, M–O**), respectively. Scale bar: 200 µm. Mes: mesenchyme; PPM: palatine process of maxilla; PB: palatine bone.

The online version of this article includes the following source data and figure supplement(s) for figure 4:

**Source data 1.** Source data for *Figure 4C*.

**Source data 2.** Source data for *Figure 4F*.

**Figure supplement 1.** *Trp53* plays a critical role in regulating palatogenesis.

**Figure supplement 1—source data 1.** Source data for *Figure 4—figure supplement 1A*.

**Figure supplement 1—source data 2.** Source data for *Figure 4—figure supplement 1B*.

**Figure supplement 1—source data 3.** Source data for *Figure 4—figure supplement 1C*.

*Figure 4 continued on next page*

*Figure 4 continued*

**Figure supplement 1—source data 4.** Source data for *Figure 4—figure supplement 1F*.
**Figure supplement 1—source data 5.** Source data for *Figure 4—figure supplement 1G*.

out of five *Wnt1^Cre^;Kdm6b^fl/fl^* mice (**Figure 4G–R**). The remaining two showed a normal hard palate, but presented with posterior soft palate defects. Western blot analysis showed that the protein level of P53 was successfully restored in the Nutlin-3-treated group (**Figure 4—figure supplement 1F and G**). This result further revealed that downregulation of *Trp53* in *Wnt1^Cre^;Kdm6b^fl/fl^* mice plays an essential role in the palatal defects. The genetic interaction between *Kdm6b* and *Trp53* is important for the development of post-migratory CNCCs.

## Level of H3K27me3 is antagonistically regulated by *Kdm6b* and *Ezh2* during palatogenesis

The lysine-specific demethylase KDM6B is able to activate gene expression via removing the H3K27me3 repressive mark (**Jiang et al., 2013**). To investigate whether *Kdm6b* regulates the expression of *Trp53* through modifying the level of H3K27me3, we first examined the status of H3K27me3 in our samples and found that loss of *Kdm6b* in CNC-derived cells resulted in accumulation of H3K27me3 in the nucleus of CNC-derived palatal mesenchymal cells (**Figure 5A–D**). Furthermore, immunoblotting revealed that the level of H3K27me3 was increased in the palatal region of *Kdm6b* mutant mice (**Figure 5E**). Since the level of H3K27me3 can also be modified by the methyltransferases EZH1 and EZH2, we further evaluated whether expression of EZH1 and EZH2 was affected in the palatal region. We found no obvious differences in either the distribution of *Ezh1+* cells or the EZH1 protein level between control and *Kdm6b* mutant mice (**Figure 5F–J**). Similarly, no dramatic changes were observed in either the distribution of EZH2+ cells or its protein level between control and *Kdm6b* mutant mice (**Figure 5K–O**). These results indicated that increased H3K27me3 in *Wnt1^Cre^;Kdm6b^fl/fl^* mice was mainly caused by loss of *Kdm6b* in CNC-derived cells. However, we did notice a broader contribution and stronger signal of EZH2 than EZH1 in the CNC-derived palatal mesenchyme. To investigate whether an increase of H3K27me3 in the CNC-derived cells caused the cleft phenotype we observed in *Wnt1^Cre^;Kdm6b^fl/fl^* mice, we generated *Wnt1^Cre^;Kdm6b^fl/fl^;Ezh2^fl/+^* mice and assessed the level of H3K27me3 in this model. In *Wnt1^Cre^;Kdm6b^fl/fl^;Ezh2^fl/+^* mice, we observed a rescue of the abnormal accumulation of H3K27me3 (**Figure 5P–V**). More importantly, haploinsufficiency of *Ezh2* in this model successfully rescued the cleft palate phenotype observed in *Wnt1^Cre^;Kdm6b^fl/fl^* mice (**Figure 6A–O**) with 70% efficiency (N = 10). CT scanning showed that both the palatine processes of the maxilla and palatine bone were restored in the *Wnt1^Cre^;Kdm6b^fl/fl^;Ezh2^fl/+^* mice (**Figure 6A–C**). Both bone and palatal mesenchymal tissue were rescued in *Wnt1^Cre^;Kdm6b^fl/fl^;Ezh2^fl/+^* mice (**Figure 6D–O**). Furthermore, both EdU+ and RUNX2+ cells were restored to normal levels in *Wnt1^Cre^;Kdm6b^fl/fl^;Ezh2^fl/+^* mice (**Figure 7A–H**). These results suggested that an antagonistic interaction between the histone demethylase KDM6B and methyltransferase EZH2 that modulates H3K27me3 is essential for palatogenesis.

## *Kdm6b* activates expression of *Trp53* through removing H3K27me3 at the promoter of *Trp53* and interacts with transcription factor TFDP1 in regulating P53 signaling pathway

Chromatin accessibility represents the degree to which chromatinized DNA is able to physically interact with nuclear macromolecules such as transcription factors for gene regulation (**Klemm et al., 2019**). The repressive mark H3K27me3 is usually associated with facultative heterochromatin and results in transcriptional repression due to decreased chromatin accessibility (**Wiles and Selker, 2017**; **Möller et al., 2019**; **den Broeder et al., 2020**). The methyltransferase EZH2 and demethylases KDM6A/KDM6B can regulate the methylation status of H3K27 to affect gene expression (**Pediconi et al., 2019**). To test whether *Kdm6b* and *Ezh2* can regulate expression of *Trp53* via H3K27me3, we first examined whether deposition of H3K27me3 changes at the promoter of *Trp53* in our models using ChIP-qPCR. A primer set was designed at 1127 bp upstream of *Trp53* exon 1, and the results showed that deposition of H3K27me3 significantly increased at the promoter of *Trp53* in the palatal region of *Kdm6b* mutant mice, while this increase was dampened in the *Ezh2* haploinsufficiency

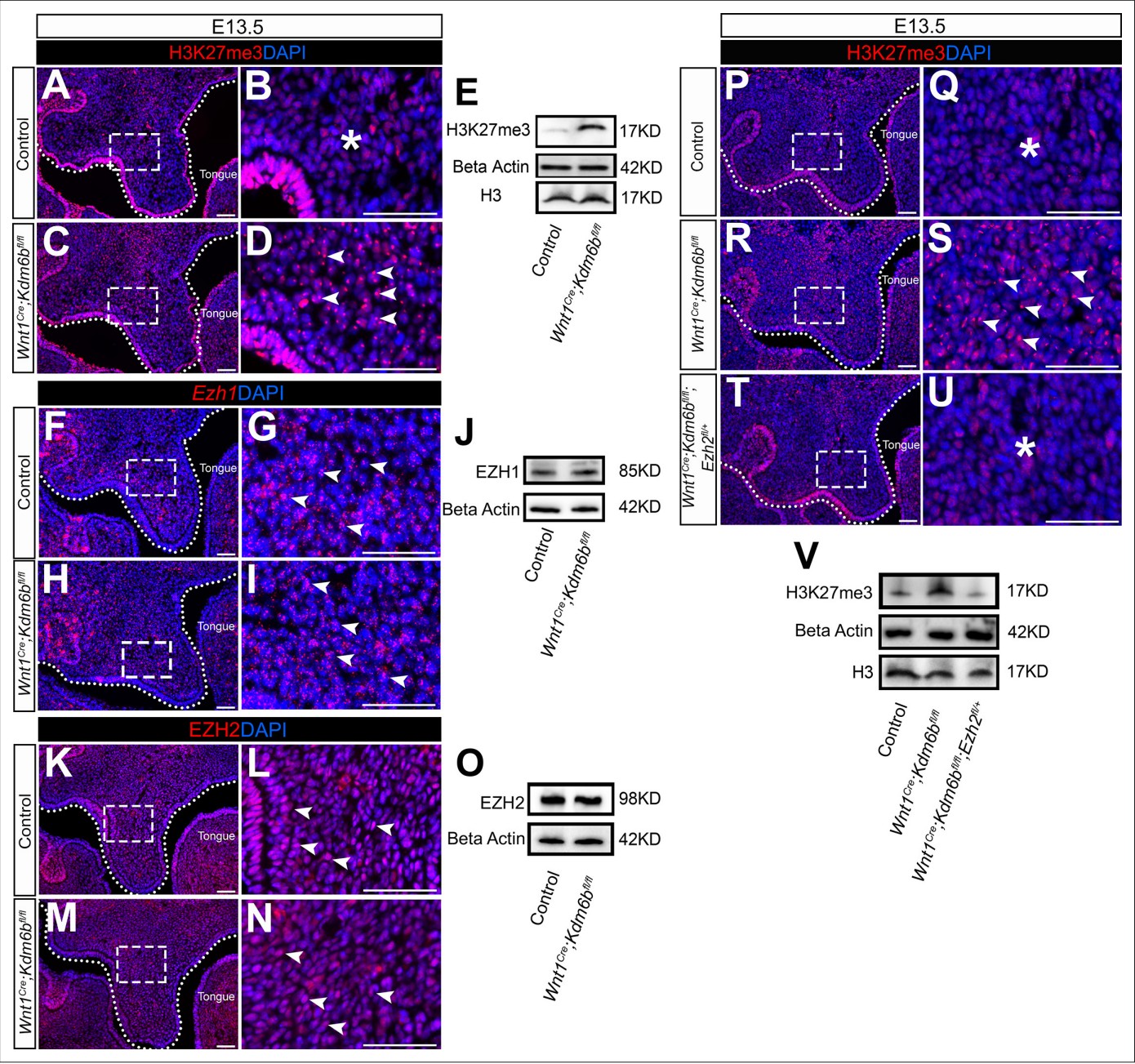

**Figure 5.** Level of H3K27me3 is antagonistically regulated by *Kdm6b* and *Ezh2* during palatogenesis. (**A–E**) Contribution of H3K27me3 in the palatal shelf is evaluated using immunostaining and Western blot at E13.5. Dotted lines in (**A, C**) indicate palatal shelf region. (**B, D**) are magnified images of boxes in (**A, C**). Asterisk in (**B**) indicates no accumulation of H3K27me3 observed in control mice. Arrowheads in (**D**) indicate accumulation of H3K27me3 observed in *Wnt1^Cre^;Kdm6b^fl/fl^* mice. Scale bar: 50 μm. (**F–J**) Contribution of *Ezh1* in the palatal shelf is evaluated using RNAscope in situ hybridization and Western blot at E13.5. Dotted lines in (**F, H**) indicate palatal shelf region. (**G, I**) are magnified images of boxes in (**F, H**). Arrowheads in (**G, I**) indicate representative *Ezh1+* cells. Scale bar: 50 μm. (**K–O**) Contribution of EZH2 in the palatal shelf is evaluated using immunostaining and Western blot at E13.5. Dotted lines in (**K, M**) indicate palatal shelf region. (**L, N**) are magnified images of boxes in (**K, M**). Arrowheads in (**L, N**) indicate representative EZH2+ cells. Scale bar: 50 μm. (**P–V**) Contribution of H3K27me3 in the palatal shelf of control mice, *Kdm6b* mutant mice, and EZH2 haploinsufficient model is evaluated using immunostaining and Western blot at E13.5. Dotted lines in (**P, R, T**) indicate palatal shelf region. (**Q, S, U**) are magnified images of boxes in (**P, R, T**), respectively. Asterisks in (**Q, U**) indicate no accumulation of H3K27me3 observed in control (**Q**) and *Wnt1^Cre^;Kdm6b^fl/fl^;Ezh2^fl/+^* mice (**U**). White arrowheads in (**S**) indicate accumulation of H3K27me3 observed in *Wnt1^Cre^;Kdm6b^fl/fl^* mice. Scale bar: 50 μm.

The online version of this article includes the following source data for figure 5:

**Source data 1.** Source data for *Figure 5E*.

*Figure 5 continued on next page*

*Figure 5 continued*
**Source data 2.** Source data for *Figure 5J*.
**Source data 3.** Source data for *Figure 5O*.
**Source data 4.** Source data for *Figure 5V*.

model (*Figure 8A*). Meanwhile, haplosufficiency of *Ezh2* in *Wnt1^Cre^;Kdm6b^fl/fl^;Ezh2^fl/+^* mice was able to restore the decreased expression of *Trp53* observed in the CNC-derived palatal mesenchyme of *Wnt1^Cre^;Kdm6b^fl/fl^* mice (*Figure 8B–H*). These data suggested that *Kdm6b* and *Ezh2* co-regulate expression of *Trp53* through H3K27me3. To further reveal whether KDM6B regulates the expression of *Trp53* directly, we performed ChIP-qPCR using KDM6B antibody and found that deposition of KDM6B significantly increased at the promoter of *Trp53* in the palatal region (*Figure 8I*). In addition, to test whether KDM6B has a unique role in activating *Trp53* during palatogenesis, we transfected palatal mesenchymal cells from *Kdm6b* mutant mice with either a plasmid overexpressing *Kdm6b* or a plasmid overexpressing another histone demethylase, *Kdm6a*. Increased expression of *Trp53* could be detected only in the group transfected with *Kdm6b*-overexpressing plasmid (*Figure 8J and K*, *Figure 8—figure supplement 1A and B*). These results indicated that *Kdm6b* has an essential and unique role in activating *Trp53* during palatogenesis.

As an H3K27me3 demethylase, KDM6B is important for the regulation of chromatin structure for gene expression. To target a specific sequence in genome, a histone demethylase needs to interact with DNA binding proteins such as transcription factors or lncRNAs (*Dimitrova et al., 2015*; *Gurrion et al., 2017*). To identify a transcription factor that can interact with KDM6B, we performed ATAC-seq analysis of palate tissue at E13.5. Through motif analysis, we found that the promoter region of *Trp53* was accessible to members of the E2F transcription factor family (E2F4 and E2F6) and transcription factor TFDP1 (also known as Dp1), a binding partner of E2F family members (*Figure 8L*, *Figure 8—figure supplement 1C*). We noticed that this open region was also located at the Transcription start site (TSS) of *Wrap53*, which was previously reported to regulate endogenous *Trp53* mRNA levels and P53 protein levels (*Mahmoudi et al., 2009*). To evaluate whether the decrease of *Trp53* was caused by altered expression of *Wrap53* in our model, we examined the expression of *Wrap53* in the palatal region at E13.5. Almost no expression of *Wrap53* was detected in the palatal region (*Figure 8—figure supplement 1D and E*). Previous research reported that inactivation of E2Fs resulted in milder phenotypes than those associated with loss of *Tfdp1*, which leads to early embryonic lethality (*Kohn et al., 2003*). This result suggested that TFDP1 may play a more critical role than E2Fs during embryonic development. A motif of TFDP1 was detected 1011 bp upstream of *Trp53* exon 1, which is very close to the H3K27me3 deposition site, by ATAC-seq analysis. ChIP-qPCR using palate tissue at E13.5 also revealed that binding of TFDP1 to the promoter region of *Trp53* decreased in the *Kdm6b* mutant mice (*Figure 8M*). Immunohistochemistry analysis showed that TFDP1+ cells were distributed in the palatal region and co-expressed with *Trp53* and *Kdm6b* (*Figure 8N and O*). We also confirmed co-expression of *Kdm6b* and *Tfdp1* in the palatal region (especially enriched in *Pax9*+, *Aldh1a2*+, and *Twist1*+ cells) using our previously published scRNA-seq data (*Figure 8—figure supplement 1F*; *Han et al., 2021*). Meanwhile, the expression level and distribution of TFDP1 were not affected in the palatal mesenchyme of *Wnt1^Cre^;Kdm6b^fl/fl^* mice (*Figure 8P*, *Figure 8—figure supplement 1G and H*). These data indicated that *Tfdp1* is not a downstream target of *Kdm6b*. To further test whether *Tfdp1* regulated expression of *Trp53* in the palatal mesenchymal cells, we transfected palatal mesenchymal cells from control mice at E13.5 using siRNA to knock down *Tfdp1*. qPCR revealed that the expression of *Trp53* was decreased in cells treated with siRNA for *Tfdp1* (*Figure 8Q*, *Figure 8—figure supplement 1I*). This data further indicated that *Trp53* is a direct downstream target of *Tfdp1*.

To reveal the function of *Kdm6b-Tfdp1* interaction in the regulation of *Trp53* during palatogenesis, we transfected palatal mesenchymal cells with *Tfdp1*-overexpressing plasmid and found that the expression of *Trp53* increased in the cells from control mice but not in the cells from *Wnt1^Cre^;Kdm6b^fl/fl^* mice (*Figure 8R and S*, *Figure 8—figure supplement 1J and K*). This result suggested that *Kdm6b* is specifically required and plays an essential role in the activation of *Trp53* through interaction with *Tfdp1* during palatogenesis. We performed co-immunoprecipitation (Co-IP) experiments and found that KDM6B and TFDP1 were indeed involved in the same complex (*Figure 8T*). Collectively, these

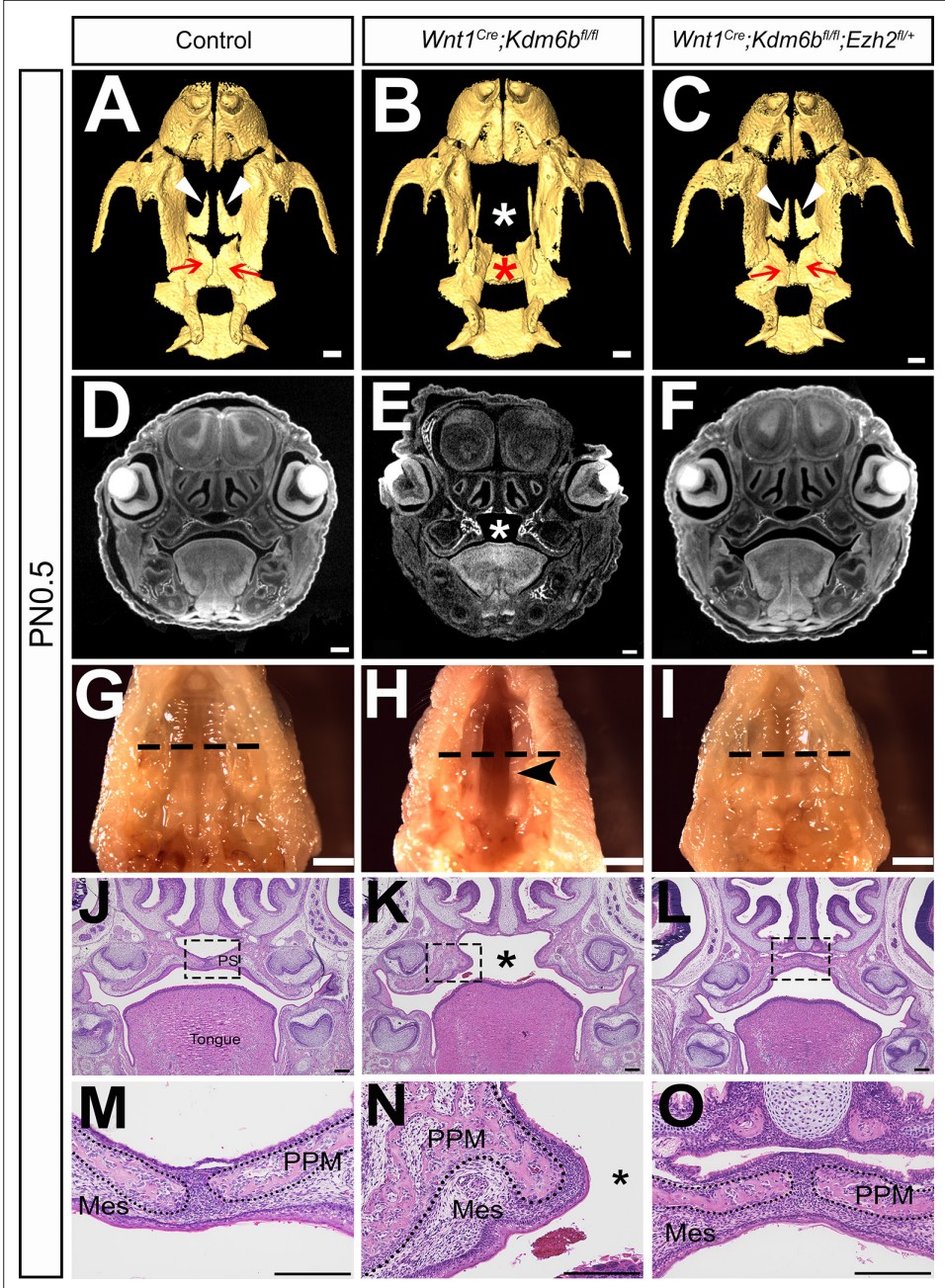

**Figure 6.** Haploinsufficiency of *Ezh2* rescues cleft palate in *Wnt1^Cre^;Kdm6b^fl/fl^;Ezh2^fl/+^* mice. (**A–C**) CT images at PN0.5. White arrowheads in (**A, C**) indicate palatine process of maxilla (PPM) observed in control and *Wnt1^Cre^;Kdm6b^fl/fl^;Ezh2^fl/+^* rescue model. Red arrows in (**A, C**) indicate palatine bone observed in control and *Wnt1^Cre^;Kdm6b^fl/fl^;Ezh2^fl/+^* rescue model. White asterisk in (**B**) indicates missing palate PPM in *Wnt1^Cre^;Kdm6b^fl/fl^* mice, and red asterisk indicates missing palatine bone in *Kdm6b* mutant mice. Scale bar: 0.4 mm. (**D–F**) Coronal views of CT images at PN0.5. Asterisk in (**E**) indicates cleft palate observed in *Wnt1^Cre^;Kdm6b^fl/fl^* mice. Scale bar: 0.3 mm. (**G–I**) Whole-mount oral view at PN0.5. Arrowhead in (**H**) shows complete cleft palate observed in *Wnt1^Cre^;Kdm6b^fl/fl^* mice. Dashed lines in (**G–I**) indicate location of sections in (**J–O**). Scale bar: 2 mm. (**J–O**) Histological analysis of samples at PN0.5. Asterisk in (**K, N**) indicates cleft palate in *Wnt1^Cre^;Kdm6b^fl/fl^* mice. (**M–O**) are magnified images of boxes in (**J–L**), respectively. Dotted lines in (**M–O**) outline the bone structure. Scale bar: 200 μm. Mes: mesenchyme.

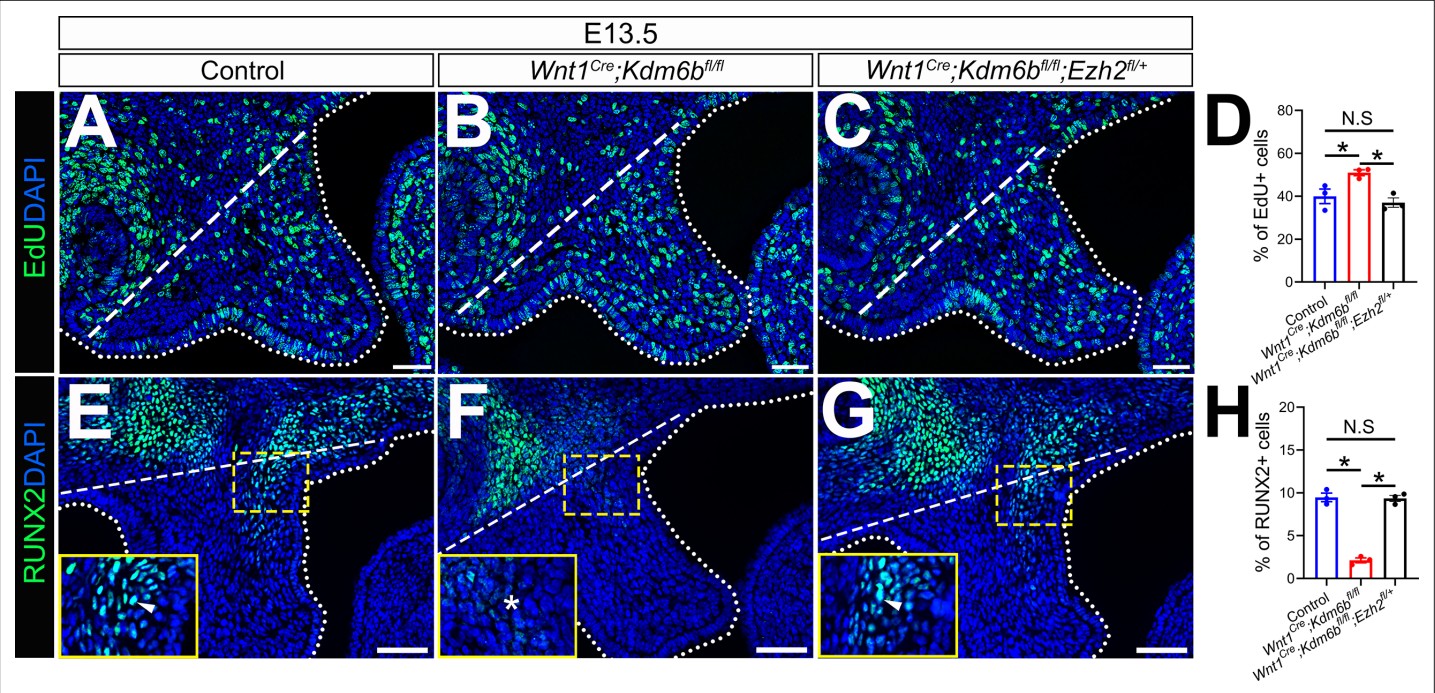

**Figure 7.** EdU+ and RUNX2+ cells are restored in *Wnt1^Cre^;Kdm6b^fl/fl^;Ezh2^fl/+^* mice. (**A–C**) Immunostaining of EdU at E13.5 after 2 hr of EdU labeling. Dotted lines indicate palatal shelf region. Dashed lines indicate the palatal region used for quantification in (**D**). Scale bar: 50 μm. (**D**) Quantification of EdU+ cells represented in (**A–C**). ANOVA is used for statistical analysis. *p<0.05. N.S: not significant. (**E–G**) Immunostaining of RUNX2 at E13.5. Dotted lines indicate palatal shelf region. Dashed lines indicate the palatal region used for quantification in (**H**). Scale bar: 50 μm. (**H**) Quantification of RUNX2+ cells represented in (**E–G**). ANOVA is used for statistical analysis. *p<0.05. N.S: not significant.

The online version of this article includes the following source data for figure 7:

**Source data 1.** Source data for *Figure 7D*.

**Source data 2.** Source data for *Figure 7H*.

data suggested that KDM6B and TFDP1 work together to activate *Trp53* expression in the palatal mesenchyme and play an important role in regulating palatogenesis (*Figure 9*).

## Discussion

The development of an organism from a single cell to multiple different cell types requires tightly regulated gene expression (*Bruneau et al., 2019*). Transcription factors, which are among the key regulators of this process, are intimately involved in cell fate commitment (*Nelms and Labosky, 2010*; *Soldatov et al., 2019*). However, a transcription factor by itself cannot act on densely packed DNA in chromatin form. Thus, transcription factors must work in coordination with epigenetic regulatory mechanisms such as histone modifications, DNA methylation, chromatin remodeling, and others to dynamically regulate chromatin states for gene expression (*Wilson and Filipp, 2018*; *Gökbuget and Blelloch, 2019*). Insults to the epigenetic landscape due to genetic, environmental, or metabolic factors can lead to diverse developmental defects and diseases (*Hobbs et al., 2014*; *Zoghbi and Beaudet, 2016*; *Flavahan et al., 2017*). Cleft palate comprises 30% of orofacial clefts and can result from genetic mutations, environmental effects, or a combination thereof (*Seelan et al., 2012*). Much progress has been made in taking inventory of the gene mutations associated with craniofacial defects in recent years, and growing evidence has shown that epigenetic regulation plays an important role during neural crest development. For example, haploinsufficiency of *KDM6A* in humans causes severe psychomotor developmental delay, global growth restriction, seizures and cleft palate (*Lindgren et al., 2013*). Furthermore, studies have shown that *Kdm6a* and *Arid1a* are both indispensable during neural crest development (*Chandler and Magnuson, 2016*; *Shpargel et al., 2017*). DNA methyl-transferase3A (DNMT3A) plays a critical role in mediating the transition from neural tube to neural

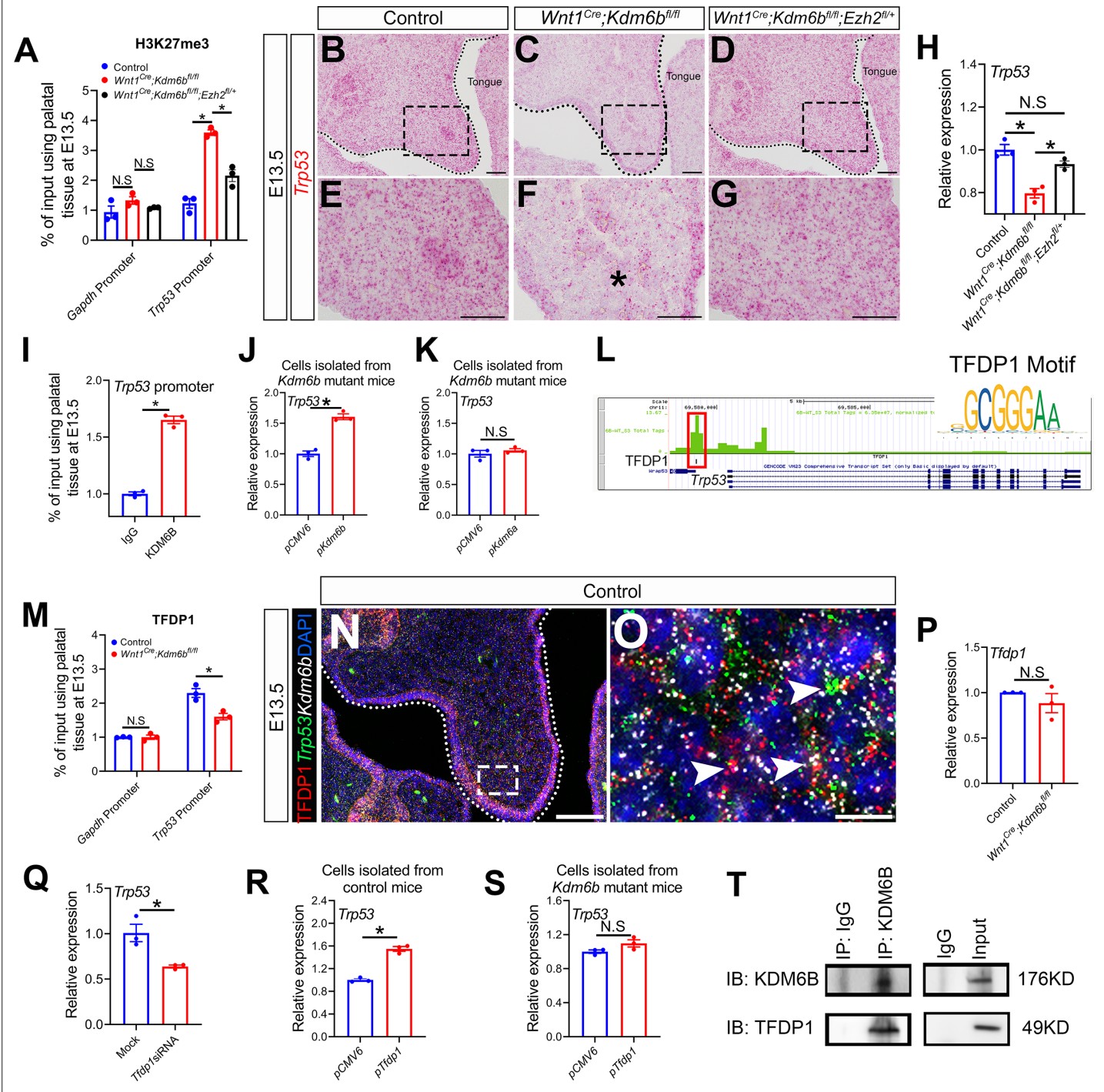

**Figure 8.** *Kdm6b* regulates expression of *Trp53* through H3K27me3 and interacts with transcription factor TFDP1 in the activation of *Trp53*. (**A**) ChIP-qPCR shows H3K27me3 deposition at the promoter region of *Trp53* in palatal tissues of control, *Kdm6b* mutant, and *Ezh2* haploinsufficient mice. ANOVA is used for statistical analysis. *p<0.05. N.S: not significant. (**B–G**) Expression of *Trp53* in the palatal region at E13.5 using RNAscope in situ hybridization. Dotted lines in (**B–D**) indicate palatal shelf. (**E–G**) are magnified images of boxes in (**B–D**), respectively. Asterisk in (**F**) indicates decreased expression of *Trp53* observed in *Wnt1^Cre^;Kdm6b^fl/fl^* mice. Scale bar: 50 μm. (**H**) RT-qPCR analysis of *Trp53* expression in the palatal region of control, *Kdm6b* mutant, and *Ezh2* haploinsufficient mice. ANOVA is used for statistical analysis. *p<0.05. N.S: not significant. (**I**) ChIP-qPCR shows KDM6B deposition at the promoter region of *Trp53* in the palatal tissue of control mice. *p<0.05. (**J, K**) RT-qPCR analysis of *Trp53* expression in palatal mesenchymal cells transfected with *Kdm6b*- or *Kdm6a*-overexpressing plasmids. *p<0.05. N.S: not significant. (**L**) ATAC-seq analysis indicates that the promoter region of *Trp53* is accessible for transcription factor TFDP1. (**M**) ChIP-qPCR using palatal tissue shows that binding of TFDP1 to the promoter of *Trp53* decreases in the *Kdm6b* mutant mice. *p<0.05. N.S: not significant. (**N, O**) Co-localization of TFDP1, *Kdm6b,* and *Trp53* at E13.5 using

*Figure 8 continued on next page*

*Figure 8 continued*

immunostaining and RNAscope in situ hybridization. Dotted lines in (**N**) indicate palatal shelf. (**O**) is a magnified image of the box in (**N**). Arrowheads in (**O**) indicate representative cells that are positive for TFDP1, *Kdm6b,* and *Trp53*. Scale bar: 50 µm in (**N**) and 5 µm in (**O**). (**P**) RT-qPCR quantification shows the expression of *Tfdp1* in samples collected at E13.5. N.S: not significant. (**Q**) RT-qPCR analysis of *Trp53* expression in palatal mesenchymal cells after *Tfdp1* siRNA transfection. *p<0.05. (**R, S**) RT-qPCR analysis of *Trp53* expression in palatal mesenchymal cells transfected with *Tfdp1* overexpressing plasmid. (**R**) *p<0.05. N.S: not significant. (**T**) Co-immunoprecipitation (Co-IP) experiment using protein extract from palatal tissues indicates that KDM6B and TFDP1 are present in the same complex. Anti-KDM6B antibody was used for immunoprecipitation (IP). IgG served as negative control. IB: immunoblotting.

The online version of this article includes the following source data and figure supplement(s) for figure 8:

**Source data 1.** Source data for *Figure 8A*.

**Source data 2.** Source data for *Figure 8H*.

**Source data 3.** Source data for *Figure 8I*.

**Source data 4.** Source data for *Figure 8J*.

**Source data 5.** Source data for *Figure 8K*.

**Source data 6.** Source data for *Figure 8M*.

**Source data 7.** Source data for *Figure 8P*.

**Source data 8.** Source data for *Figure 8Q*.

**Source data 9.** Source data for *Figure 8R*.

**Source data 10.** Source data for *Figure 8S*.

**Source data 11.** Source data for *Figure 8T*.

**Figure supplement 1.** KDM6B and transcription factors are involved in regulating *Trp53*.

**Figure supplement 1—source data 1.** Source data for *Figure 8—figure supplement 1A*.

**Figure supplement 1—source data 2.** Source data for *Figure 8—figure supplement 1B*.

**Figure supplement 1—source data 3.** Source data for *Figure 8—figure supplement 1I*.

**Figure supplement 1—source data 4.** Source data for *Figure 8—figure supplement 1J*.

**Figure supplement 1—source data 5.** Source data for *Figure 8—figure supplement 1K*.

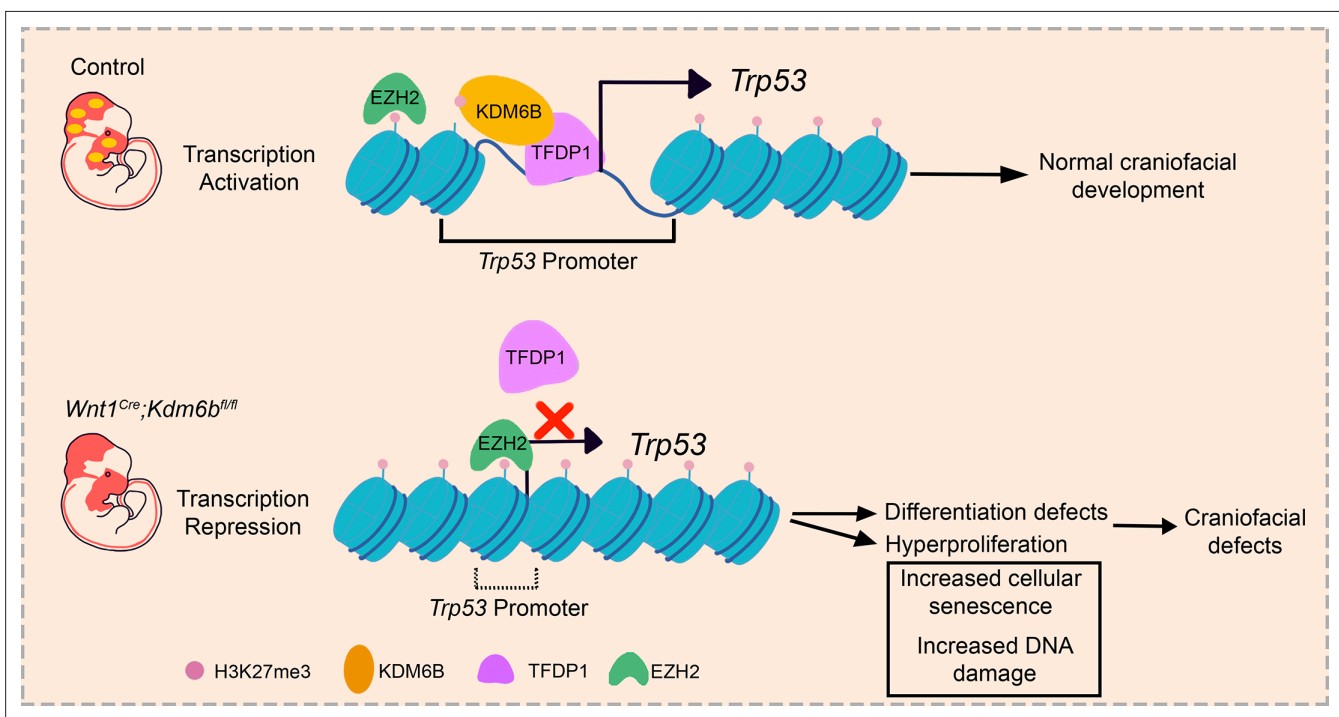

**Figure 9.** Summary schematic drawing.

crest fate (*Hu et al., 2012*). Meanwhile, loss of *Ezh2*, a component of PRC2, in CNC-derived cells completely prevents craniofacial bone and cartilage formation (*Schwarz et al., 2014*). These studies have clearly shown that epigenetic regulation is crucial for neural crest development. In this study, we further demonstrate the important role of epigenetic regulation during the neural crest contribution to palate development using *Wnt1^Cre^;Kdm6b^fl/fl^* mice as a model. We show that the demethylase KDM6B is not only required for normal CNC-derived palatal mesenchymal cell proliferation, but also for maintaining cell differentiation.

Epigenetic regulators, transcription factors, and lineage-specific genes work together to achieve spatiotemporally restricted, tissue-specific gene regulation (*Hu et al., 2014*). In this study, we reveal that *Kdm6b* works with the transcription factor *Tfdp1* to specifically regulate the expression of *Trp53* during palatogenesis. The molecular mechanisms underlying the function of *Trp53* in genomic stability and tumor suppression have been studied extensively. However, the role of *Trp53* in regulating the development of CNC-derived cells still remains largely unclear, although several studies have been conducted recently on certain aspects of this topic. For instance, it has been shown that *Trp53* is able to coordinate CNC cell growth and epithelial-mesenchymal transition/delamination processes by modulating cell cycle genes and proliferation (*Rinon et al., 2011*). It has also been established that both deletion and overexpression of *Trp53* result in craniofacial defects (*Tateossian et al., 2015*; *Bowen et al., 2019*). Furthermore, nuclear stabilization of P53 protein in *Tcof^+/-^* induces neural crest cell progenitors to undergo cell cycle arrest and caspase3-mediated apoptosis in the neuroepithelium. Inhibition of P53 function successfully rescues the neurocristopathy in an animal model of Treacher Collins syndrome, which results from mutation in *Tcof1* (*Jones et al., 2008*). These studies have clearly shown that appropriate function of *Trp53* is indispensable in CNCCs. However, none of these studies have addressed upstream regulation of *Trp53* in CNCCs.

Here, we show that proper function of *Trp53* during the differentiation and proliferation of CNCCs is orchestrated by *Kdm6b* and *Ezh2* through H3K27me3. Altering the balance between *Ezh2* and *Kdm6b* can cause abnormal H3K27me3 function, which further affects the downstream transcription factor *Trp53*. In addition, we have detected spontaneous DNA damage in the developing palate and increased accumulation of DNA damage in the *Wnt1^Cre^;Kdm6b^fl/fl^* mice. These findings further demonstrate the critical function of *Trp53* in protecting embryonic cells from DNA damage during development. Furthermore, the ability of cells to proliferate is limited by the length of the telomeres, which gradually shorten during each cell replication (*Blagoev, 2009*). Once the telomeres are too short for DNA replication, the result is cellular senescence, which induces an irreversible inability to proliferate (*Bernadotte et al., 2016*). In this study, we show that downregulated expression of *Trp53* in *Wnt1^Cre^;Kdm6b^fl/fl^* mice results in hyperproliferation and increased DNA damage in the proliferative cells, which might further lead to increased cell senescence. Previous studies reported that few *Trp53^-/-^* mice exhibit cleft palate or other craniofacial abnormalities, which suggests that loss of *Trp53* alone is not powerful enough to cause defects during craniofacial development (*Rinon et al., 2011*; *Tateossian et al., 2015*) and that there might be other factors that can compensate for the loss of function of *Trp53*. In our study, *Wnt1^Cre^;Kdm6b^fl/fl^* mice showed a cleft palate phenotype with high penetrance, and using MDM2 inhibitor Nutlin-3 we successfully rescued cleft palate in *Kdm6b* mutant mice. Our results suggested that in the *Kdm6b* mutant mice the decrease of *Trp53* cannot be compensated for by other factors, and disturbed P53 signaling is the key factor causing cleft palate in *Kdm6b* mutant mice.

Previous research has shown that KDM3A functions as a cofactor of STAT3 to activate the JAK2-STAT3 signaling pathway (*Kim et al., 2018*) and that KDM2A coordinates with c-Fos in regulating COX-*2* (*Lu et al., 2015*). Our study shows that KDM6B coordinates with the transcription factor TFDP1 to activate expression of *Trp53* in CNCCs, and that *Ezh2* and *Kdm6b* co-regulate H3K27 methylation status, which may affect the ability of TFDP1 to bind to the chromatin during palatogenesis. It has been reported that *Tfdp1* is crucial for embryonic development and regulating Wnt/β-catenin signaling (*Kohn et al., 2003*; *Kim et al., 2012*). Interaction between KDM6B and TFDP1 discovered in this study further increases our knowledge of the coordination between epigenetic regulators and transcription factors during organogenesis. As environmental insults can adversely affect the function of epigenetic regulators, our findings provide a better understanding of the epigenetic regulation and transcription factors involved in regulating the fate of CNC cells and craniofacial development, which

can provide important clues about human development, as well as potential therapeutic approaches for craniofacial birth defects.

# Materials and methods

## Key resources table

| Reagent type (species) or resource | Designation | Source or reference | Identifiers | Additional information |
|---|---|---|---|---|
| Strain, strain background (*Mus musculus*) | *Kdm6b*<sup>flox/flox</sup> | *Manna et al., 2015*, Jackson Laboratory | Stock# 029615; RRID:IMSR_JAX:029615 | |
| Strain, strain background (*M. musculus*) | Ezh2<sup>flox/flox</sup> | Jackson Laboratory | Stock# 022616; RRID:IMSR_JAX:022616 | |
| Strain, strain background (*M. musculus*) | Trp53<sup>flox/flox</sup> | Jackson Laboratory | Stock# 008462; RRID:IMSR_JAX:008462 | |
| Strain, strain background (*M. musculus*) | Wnt1<sup>Cre</sup> | *Zhao et al., 2008* | | |
| Strain, strain background (*M. musculus*) | Krt14<sup>cre</sup> | Jackson Laboratory | Stock# 018964; RRID:IMSR_JAX:018964 | |
| Strain, strain background (*M. musculus*) | ROSA26loxp-STOP-loxp-tdTomato | Jackson Laboratory | Stock# 007905; RRID:IMSR_JAX:007905 | |
| Sequence-based reagent | Mm-*Kdm6a* probe | Advanced Cell Diagnostics | Cat# 456961 | |
| Sequence-based reagent | Mm-*Kdm6b* probe | Advanced Cell Diagnostics | Cat# 477971 | |
| Sequence-based reagent | Mm-*Kdm6b*-01 probe | Advanced Cell Diagnostics | Cat# 501231 | |
| Sequence-based reagent | Mm-*Uty* probe | Advanced Cell Diagnostics | Cat# 451741 | |
| Sequence-based reagent | Mm-*Trp53* probe | Advanced Cell Diagnostics | Cat# 402331 | |
| Sequence-based reagent | Mm-*Trp53*-C2 probe | Advanced Cell Diagnostics | Cat# 402331-C2 | |
| Sequence-based reagent | Mm-*Ezh1* probe | Advanced Cell Diagnostics | Cat# 415501 | |
| Sequence-based reagent | Mm-*Wrap53* probe | Advanced Cell Diagnostics | Cat# 1143201-C1 | |
| Antibody | Myosin heavy chain (MHC) (mouse monoclonal) | DSHB | Cat# P13538 | (1:10) |
| Antibody | Histone H3 tri methyl K27 (rabbit monoclonal) | Cell Signaling Technology | Cat# 9733s | (1:200) (1:1000) |
| Antibody | Phospho-histone H2A.X (rabbit monoclonal) | Cell Signaling Technology | Cat# 9718s | (1:200) |
| Antibody | DP1 (rabbit monoclonal) | Abcam | Cat# ab124678 | (1:100) (1:1000) |
| Antibody | EZH2 (rabbit monoclonal) | Cell Signaling Technology | Cat# 5246s | (1:200) (1:2000) |
| Antibody | RUNX2 (rabbit monoclonal) | Cell Signaling Technology | Cat# 12556s | (1:200) |
| Antibody | SP7 (rabbit polyclonal) | Abcam | Cat# ab22552 | (1:200) |
| Antibody | Lamin B1 (rabbit monoclonal) | Cell Signaling Technology | Cat# 17416s | (1:100) |
| Antibody | Anti-mouse Alexa Fluor 488 (goat polyclonal) | Life Technologies | Cat# A11001 | (1:200) |
| Antibody | Anti-mouse Alexa Fluor 568 (goat polyclonal) | Life Technologies | Cat# A-11004 | (1:200) |
| Antibody | Anti-rat Alexa Fluor 488 (goat polyclonal) | Life Technologies | Cat# A-11006 | (1:200) |
| Antibody | Anti-rabbit Alexa Fluor 488 (goat polyclonal) | Life Technologies | Cat# A-11008 | (1:200) |
| Antibody | Anti-rabbit Alexa Fluor 568 (goat polyclonal) | Life Technologies | Cat# A-11036 | (1:200) |
| Antibody | EZH1 (rabbit polyclonal) | Abcam | Cat# ab189833 | (1:1000) |
| Antibody | KDM6A (rabbit polyclonal) | Abcam | Cat# ab36938 | (1:1000) |
| Antibody | KDM6B (C-term) (rabbit polyclonal) | AbCEPTA | Cat# AP1022b | (1:1000) |
| Antibody | KDM6B (N-term) (rabbit polyclonal) | AbCEPTA | Cat# AP1022a | (1:1000) |
| Antibody | P53 (mouse monoclonal) | Santa Cruz | Cat# sc-126 | (1:1000) |

*Continued on next page*

*Continued*

| Reagent type (species) or resource | Designation | Source or reference | Identifiers | Additional information |
|---|---|---|---|---|
| Antibody | Histone H3 (rabbit monoclonal) | Cell Signaling Technology | Cat# 4499s | (1:1000) |
| Antibody | β-Actin (mouse monoclonal) | Abcam | Cat# Ab20272 | (1:2000) |
| Antibody | Rabbit IgG HRP-conjugated antibody (goat polyclonal) | R&D System | Cat# HAF008 | (1:2000) |
| Antibody | Mouse IgG HRP-conjugated antibody (goat polyclonal) | R&D System | Cat# HAF007 | (1:2000) |
| Antibody | HRP, mouse anti-rabbit IgG LCS (mouse monoclonal) | IPKine | Cat# A25022 | (1:2000) |
| Commercial assay or kit | Goat anti-mouse IgG Alexa Fluor 488 Tyramide SuperBoost Kit | Thermo Fisher Scientific | Cat# B40912 | (1:200) |
| Commercial assay or kit | RNAscope Multiplex Fluorescent Kit v2 | Advanced Cell Diagnostics | Cat# 323110 | |
| Commercial assay or kit | RNAscope 2.5 HD Assay – RED | Advanced Cell Diagnostics | Cat# 322350 | |
| Commercial assay or kit | TSA Plus Cyanine 3 System | PerkinElmer | Cat# NEL744001KT | |
| Commercial assay or kit | TSA Plus Fluoresceine System | PerkinElmer | Cat# NEL771B001KT | |
| Commercial assay or kit | RNeasy Micro Kit | QIAGEN | Cat# 74004 | |
| Commercial assay or kit | DAB Peroxidase (HRP) Substrate Kit (with nickel) | Vector Laboratories | RRID:AB_2336382; Cat# SK4100 | |
| Software, algorithm | ImageJ | NIH | RRID:SCR_003070 | |
| Software, algorithm | Ingenuity Pathway Analysis | QIAGEN, Inc | RRID:SCR_008653 | |
| Software, algorithm | GraphPad Prism | GraphPad Software | RRID:SCR_002798 | |
| Software, algorithm | Seurat | Satija lab | RRID:SCR_016341 | |
| Software, algorithm | Cell Ranger | 10X Genomics, Inc | RRID:SCR_017344 | |

## Animals

To generate *Wnt1^Cre^;Kdm6b^fl/fl^* mice, we crossed *Wnt1^Cre^;Kdm6b^fl/+^* mice with *Kdm6b^fl/fl^* mice (*Zhao et al., 2008*; *Manna et al., 2015*). Reporter mice used in this study were tdTomato conditional reporter (JAX#007905) (*Madisen et al., 2010*). *Ezh2^fl/fl^* and *Trp53^fl/fl^* mice were purchased from Jackson Laboratory (JAX#022616, #008462) (*Marino et al., 2000*; *Shen et al., 2008*). Genotyping was carried out as previously described (*Zhao et al., 2008*). Briefly, tail samples were lysed by using DirectPCR tail solution (Viagen 102T) with overnight incubation at 55°C. After heat inactivation at 85°C for 1 hr, PCR-based genotyping (GoTaq Green MasterMix, Promega, and C1000 Touch Cycler, Bio-Rad) was used to detect the genes. All mouse studies were conducted with protocols approved by the Department of Animal Resources and the Institutional Animal Care and Use Committee (IACUC) of the University of Southern California (Protocols 9320 and 20299).

## MicroCT analysis

MicroCT was used to analyze the control, *Kdm6b,* and other mutant samples. Mouse samples were dissected and fixed in 4% paraformaldehyde overnight at 4°C followed by CT scanning (Scanco Medical µCT50 scanner) at the University of Southern California Molecular Imaging Center as previously described (*Grosshans et al., 2006*; *Sugii et al., 2017*). AVIZO 9.1.0 (Visualization Sciences Group) was used for visualization and 3D microCT reconstruction.

## Alcian blue-Alizarin red staining

Mouse heads were dissected and fixed in 95% EtOH overnight at room temperature. Staining was performed as previously described (*Rigueur and Lyons, 2014*). Briefly, 95% EtOH was replaced with 100% acetone for 2 days and then samples were incubated in Alcian blue solution (80% EtOH, 20% glacial acetic acid, and 0.03% [w/v] Alcian blue 8GX [Sigma, A3157]) for 1–3 days. Samples were then de-stained with 70% EtOH and incubated in 95% EtOH overnight. After incubation, samples were pre-cleared with 1% KOH and then incubated in Alizarin red solution (0.005% [w/v] Alizarin red [Sigma,

A5533] in 1% [w/v] KOH) for 2–5 days. After clearing samples with 1% KOH, they were stored in 100% glycerol until analysis.

## Sample preparation for sectioning

Samples for paraffin sectioning were prepared using the standard protocol in our laboratory. Briefly, samples were fixed in 4% PFA and decalcified with 10% EDTA as needed. Then, the samples were dehydrated with serial ethanol solutions (50, 70, 80, 90, and 100%) at room temperature followed by xylene and then embedded in paraffin wax. Sections were cut to 6 µm on a microtome (Leica) and mounted on SuperFrost Plus slides (Fisher, 48311-703). Cryosectioning samples were fixed and decalcified the same way as samples prepared for paraffin sectioning. Sucrose (15 and 30%) was used to remove water from the samples before embedding them in OCT compound (Tissue-Tek, 4583). Cryosections were cut to 8 µm on a cryostat (Leica) and mounted on SuperFrost Plus slides (Fisher).

## Histological analysis

Paraffin sections prepared as described above were used for histological analysis. Hematoxylin and eosin staining was performed using the standard protocol (*Cardiff et al., 2014*).

## Immunofluorescence assay

Cryosections and paraffin sections prepared as described above were used for immunofluorescence assays. Sections were dried for 2 hr at 55°C. Paraffin sections were deparaffinized and rehydrated before antigen retrieval. Heat-mediated antigen retrieval was used to process sections (Vector, H-3300) and then samples were blocked for 1 hr in blocking buffer at room temperature (PerkinElmer, FP1020). Primary antibodies diluted in blocking buffer were incubated with samples overnight at 4°C. After washing with PBST (0.1% Tween20 in 1× PBS), samples were then incubated with secondary antibodies at room temperature for 2 hr. DAPI (Sigma, D9542) was used for nuclear staining. All images were acquired using Leica DMI 3000B and Keyence BZ-X710/810 microscopes. Detailed information about the primary and secondary antibodies is listed in *Supplementary file 1*.

## EdU labeling

EdU solution was prepared at 10 mg/mL in PBS, and then pregnant mice at the desired stage were given an intraperitoneal injection (IP) based on their weight (0.1 mg of EdU/1 g of mouse). Embryos were collected after 2 hr or 48 hr and then prepared for sectioning as above. EdU signal was detected using Click-It EdU cell proliferation kit (Invitrogen, C10337), and images were acquired using Leica DMI 3000B and Keyence BZ-X710/810 microscopes.

## RNAscope in situ hybridization

RNAscope in situ hybridization in this study was performed on cryosections using RNAscope 2.5HD Reagent Kit-RED assay (Advanced Cell Diagnostics, 322350) and RNAscope multiplex fluorescent v2 assay (Advanced Cell Diagnostics, 323100) according to the manufacturer's protocol. RNAscope probes used in this study included *Kdm6a*, *Kdm6b*, *Uty*, and *Trp53*. Detailed information about the probes is listed in *Supplementary file 2*.

## RNA-sequencing and analysis

Palate samples from control and *Wnt1^{Cre}*;*Kdm6b^{fl/fl}* mice were collected at E12.5 for RNA isolation with RNeasy Micro Kit (QIAGEN) according to the manufacturer's protocol. The quality of RNA samples was determined using an Agilent 2100 Bioanalyzer, and all samples for sequencing had RNA integrity (RIN) numbers > 9. cDNA library preparation and sequencing were performed at the USC Molecular Genomics Core. Single-end reads with 75 cycles were performed on Illumina HiSeq 4000 equipment, and raw reads were trimmed and aligned using TopHat (version 2.0.8) with the mm10 genome. CPM was used to normalize the data, and differential expression was calculated by selecting transcripts that changed with $p < 0.05$.

## RNA extraction and real-time qPCR

Palatal tissue used for RNA isolation was dissected at desired stages, and an RNeasy Plus Micro Kit (QIAGEN, 74034) was used to isolate the total RNA followed by cDNA synthesis using an iScript cDNA

synthesis kit (Bio-Rad, 1708891). Real-time qPCR quantification was done on a Bio-Rad CFX96 Real-Time system using SsoFast EvaGreen Supermix (Bio-Rad, 1725201). Detailed information about the primers is listed in *Supplementary file 3*.

## ChIP-qPCR

Palate tissue was dissected from control and *Wnt1*^Cre^*;Kdm6b*^fl/fl^ mice at E13.5. Each replicate contained 60–80 mg tissue combined from multiple animals. Samples were prepared following the manufacturer's protocol (Chromatrap, 500191). Briefly, tissue was cut into small pieces and then fixed with 1% formaldehyde at room temperature for 15 min, followed by incubating with 0.65 M glycine solution. Then, the sample was washed twice with PBS, resuspended in Hypotonic Buffer, and incubated at 4°C for 10 min to obtain nuclei, which were then resuspended in Digestion Buffer. After chromatin was sheared to 100–500 bp fragments using Shearing Cocktail, 10 µg chromatin with H3K27me3 antibody (CST 9733s, 1:50), DP1 antibody (Abcam ab124678, 1:10), KDM6B antibody (Abcepta AP1022b, 1:10), or immunoglobulin G-negative control (2 µg) was added to Column Conditioning Buffer to make up the final volume of 1000 µL. Immunoprecipitation (IP) slurry was mixed thoroughly and incubated on an end-to-end rotor for 1 hr at 4°C. An equivalent amount of chromatin was set as an input. After 1 hr incubation, IP slurry was purified using Chromatrap spin column at room temperature and chromatin was eluted using ChIP-seq elution buffer. Chromatin sample and input were further incubated at 65°C overnight to reverse cross-linking. DNA was purified with Chromatrap DNA purification column after proteinase K treatment. ChIP eluates, negative control, and input were assayed using real-time qPCR. Primers were designed using the promoter region of *Trp53*. Detailed information is available in *Supplementary file 3*.

## Western blot and co-immunoprecipitation

For Western blot, palate tissue was dissected from control and *Wnt1*^Cre^*;Kdm6b*^fl/fl^ mice at E13.5. The tissue sample was lysed using RIPA buffer (Cell Signaling, 9806) with protease inhibitor (Thermo Fisher Scientific, A32929) for 20 min on ice followed by centrifugation at 4°C to remove tissue debris. Protein extracts were then mixed with sample buffer (Bio-Rad, 1610747) and boiled at 98°C for 10 min. Then, denatured protein extract was separated in 4–15% precast polyacrylamide gel (Bio-Rad, 456-1084) and then transferred to 0.45 µm PVDF membrane. Transferred membrane was incubated with 5% milk for 1 hr at room temperature and incubated with primary antibody at 4°C overnight. After washing with TBST, the membrane was incubated with secondary antibody for 2 hr at room temperature and signals were detected using SuperSignal West Femto (Thermo Fisher Scientific, 34094) and Azure 300 (Azure Biosystems).

For Co-IP, palate tissue was dissected from control mice at E13.5 and 60–80 mg tissue was combined as one sample for each replicate. After lysing using RIPA buffer, 60 µL of the protein extract was mixed with sample buffer and boiled at 98°C to serve as input. The remaining protein extract was incubated with primary antibody at 4°C overnight. Protein G beads from GE Healthcare (GE Healthcare, 10280243) were used to purify the target protein, and then the protein sample was analyzed using Western blot. Detailed information about the primary and secondary antibodies is listed in *Supplementary file 4*.

## siRNA and plasmid transfection

Palatal tissue was dissected from control and *Wnt1*^Cre^*;Kdm6b*^fl/fl^ mice at E13.5, then cut into small pieces using a scalpel. This minced tissue was then cultured in DMEM medium (Gibco, 2192449) containing 40% MSC FBS (Gibco, 2226685P) and 1% Pen Strep (Gibco, 2145477) at 37°C.

siRNA (QIAGEN) and plasmid (OriGene) transfection was performed following the manufacturer's protocol (QIAGEN, 301704, and OriGene, TF81001). Briefly, siRNA was transfected into cells in 24-well plates at 10 nM for 3 days followed by qPCR and EdU proliferation assay. Plasmid was transfected into cells in 24-well plates using 1 µg/µL stock solution for 2 days followed by real-time qPCR. The primers designed for qPCR are listed in *Supplementary file 3*. The siRNA sequence and plasmid information are listed in *Supplementary files 5 and 6*.

## ATAC-seq analysis

Palate tissue of E13.5 control mice was digested using TrypLE express enzyme (Thermo Fisher Scientific, 12605010) and incubated at 37°C for 20 min with shaking at 600 rpm. Single-cell suspension was

prepared according to the 10X Genomics sample preparation protocol and processed to generate ATAC-seq libraries according to a published protocol (*Buenrostro et al., 2015*). Sequencing was performed using the NextSeq 500 platform (Illumina), and ATAC-seq reads were aligned to the UCSC mm10 reference genome using BWA-MEN (*Li, 2013*). Then, ATAC-seq peaks were called by MACS2 and annotated. Known transcription factor biding motifs were analyzed by HOMER (*Zhang et al., 2008*; *Heinz et al., 2010*). Quality files for sequencing are listed in *Supplementary file 7*.

## Cell differentiation assay

Palatal tissue was dissected from control and *Wnt1^Cre^;Kdm6b^fl/fl^* mice at E13.5 and cultured as previously described. Then, the differentiation assay was conducted according to the manufacturer's protocol (Gibco, A1007201) (*Chen et al., 2020*). Briefly, mesenchymal cells were seeded into cell culture plates at the desired concentration followed by incubation at 36°C in a humidified atmosphere of 5% $CO_2$ for the required time (a minimum of 2 hr and up to 4 days). Then, the growth medium was replaced by complete differentiation medium and cells were continuously incubated for 3 weeks under osteogenic conditions. After specific periods of cultivation, cells were stained using 2% Alizarin red S solution (PH 4.2) solution. Images were acquired using EPSON Scan and Keyence BZ-X710/810 microscopes. Quantification of the Alizarin red S staining was conducted according to the manufacturer's protocol (ScienCell, 8678).

## Senescence β-galactosidase staining

Palatal tissue was dissected from control mice at E13.5, then cut into small pieces using a scalpel. This minced tissue was then cultured in DMEM medium (Gibco, 2192449) containing 40% MSC FBS (Gibco, 2226685P) and 1% Pen Strep (Gibco, 2145477) at 37°C. A cell monolayer was stained using a senescence β-galactosidase staining kit (Cell Signaling, 9860) according to the manufacturer's protocol. Images were acquired using a Keyence BZ-X710/810 microscope.

## Nutlin-3 treatment

Nutlin-3 (Sigma, N6287) was dissolved in corn oil (Sigma, C8267) with 10% DMSO (Sigma, D2650) and given to pregnant mice on days 10.5, 12.5, and 14.5 of pregnancy at a dosage based on their weight (10 mg/kg) (*Li et al., 2016*). Then, the embryos were collected at E16.5 for analysis.

## Statistics

Statistical analysis was completed using GraphPad Prism, and significance was assessed by independent two-tailed Student's *t*-test or ANOVA. The chosen level of significance for all statistical tests in this study was $p < 0.05$. Data is presented as mean ± SEM. N = 3 samples were analyzed for each experimental group unless otherwise stated.

# Acknowledgements

We thank Bridget Samuels and Linda Hattemer for critical reading of the manuscript and also acknowledge USC Libraries Bioinformatics Service for their assistance with data analysis. We also thank the USC Office of Research and the Norris Medical Library for the bioinformatics software and computing resources.

# Additional information

### Competing interests

Jian Xu: Reviewing editor, *eLife*. The other authors declare that no competing interests exist.

### Funding

| Funder | Grant reference number | Author |
|---|---|---|
| National Institutes of Health | R01 DE012711 | Yang Chai |

| Funder | Grant reference number | Author |
|--------|------------------------|--------|
| National Institutes of Health | R01 DE022503 | Yang Chai |
| National Institutes of Health | U01 DE028729 | Yang Chai |

The funders had no role in study design, data collection and interpretation, or the decision to submit the work for publication.

## Author contributions

Tingwei Guo, Conceptualization, Data curation, Formal analysis, Investigation, Methodology, Validation, Writing - original draft, Writing - review and editing; Xia Han, Data curation, Formal analysis, Methodology, Validation, Writing - review and editing; Jinzhi He, Data curation, Methodology, Validation, Writing - review and editing; Jifan Feng, Junjun Jing, Formal analysis, Methodology, Writing - review and editing; Eva Janečková, Jie Lei, Writing - review and editing; Thach-Vu Ho, Data curation, Formal analysis; Jian Xu, Conceptualization, Writing - review and editing; Yang Chai, Conceptualization, Data curation, Investigation, Methodology, Project administration, Resources, Supervision, Validation, Visualization, Writing - original draft, Writing - review and editing

## Author ORCIDs

Tingwei Guo ![ORCID] http://orcid.org/0000-0003-4089-1867
Jifan Feng ![ORCID] http://orcid.org/0000-0002-9944-2604
Junjun Jing ![ORCID] http://orcid.org/0000-0001-5745-5207
Thach-Vu Ho ![ORCID] http://orcid.org/0000-0001-6293-4739
Jian Xu ![ORCID] http://orcid.org/0000-0002-8162-889X
Yang Chai ![ORCID] http://orcid.org/0000-0003-2477-7247

## Ethics

All mouse studies were conducted with protocols approved by the Department of Animal Resources and the Institutional Animal Care and Use Committee (IACUC) of the University of Southern California (Protocols 9320 and 20299).

## Decision letter and Author response

Decision letter https://doi.org/10.7554/eLife.74595.sa1
Author response https://doi.org/10.7554/eLife.74595.sa2

# Additional files

## Supplementary files

- Supplementary file 1. Antibodies used for in vivo immunostaining.
- Supplementary file 2. Probes used for in situ RNAscope.
- Supplementary file 3. Primers used for ChIP-qPCR and RT-qPCR.
- Supplementary file 4. Antibodies used for Western blot and co-immunoprecipitation (co-IP).
- Supplementary file 5. siRNA used in cell culture experiments.
- Supplementary file 6. Plasmids used in cell culture experiments.
- Supplementary file 7. QC report for ATAC-seq.
- Transparent reporting form

## Data availability

Sequencing data have been deposited in GEO under accession code GSE175383.

The following dataset was generated:

| Author(s) | Year | Dataset title | Dataset URL | Database and Identifier |
|---|---|---|---|---|
| Guo T, Han X, He J, Jing J, Lei J, T-V Ho, Xu J, Chai Y, Fng J, Janeekova E | 2022 | KDM6B interacts with TFDP1 to activate P53 signalling in regulating mouse palatogenesis | http://www.ncbi.nlm.nih.gov/geo/query/acc.cgi?acc=GSE175383 | NCBI Gene Expression Omnibus, GSE175383 |

The following previously published dataset was used:

| Author(s) | Year | Dataset title | Dataset URL | Database and Identifier |
|---|---|---|---|---|
| Han X, Feng J, Guo T, Loh Y-H, Yuan Y, T-V Ho, Cho CK, Li J, Jing J, Janeckova E, He J, Pei F, Bi J, Song B, Chai Y | 2021 | Runx2-Twist1 interaction coordinates cranial neural crest guidance of soft palate myogenesis | https://www.ncbi.nlm.nih.gov/geo/query/acc.cgi?acc=GSE155928 | NCBI Gene Expression Omnibus, GSE155928 |

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
