## [Editor Report]

Using the mouse secondary palate as a model, this study reports original findings on the function of Kdm6b, a H3K27me3 demethylase, in the regulation of embryonic development. The authors show that Kdm6b plays an essential role in neural crest development, and that loss of Kdm6b perturbs the p53 pathway, leading to complete clefting of the secondary palate along with cell proliferation and differentiation defects.

---

## [Decision Letter]

**Decision letter after peer review:**

Thank you for submitting your article "Kdm6b confers Tfdp1 with the competence to activate p53 signalling in regulating palatogenesis" for consideration by *eLife*. Your article has been reviewed by 3 peer reviewers, and the evaluation has been overseen by Marianne Bronner as the Senior Editor and Reviewing Editor. The following individual involved in review of your submission has agreed to reveal their identity: Eric Liao (Reviewer #2).

Essential revisions:

The reviewers agree that this is an interesting paper with the wealth of high quality data/ However, there are several areas that need further experimentation and clarification.

1) The results as written do not fully address the stated major aim of the study, i.e. investigating "how epigenetic regulators coordinate with tissue-specific regulatory factors during morphogenesis of specific organs". This point is overemphasized in the manuscript but not fully developed so should be toned down. As case in point, the abstract overstates and goes well beyond the data presented.

2) The regulations of the p53 pathway requires additional and deeper investigations plus added discussion.

3) Additional quantitative analyses are required to support claims regarding increased numbers of apoptotic cells and decreased numbers of Runx2 positive cells in mutant palate mesenchyme.

*Reviewer #1 (Recommendations for the authors):*

Kdm6b encodes a histone demethylase that specifically demethylates H3K27me3, a repressive histone mark, and thus plays important roles in transcriptional activation of gene expression. Previous studies have shown that mice homozygous for germline loss of Kdm6b function die perinatally, but whether Kdm6b plays an important role in palate development has not been documented. In this manuscript, the showed that the palatal mesenchyme of the Wnt1-Cre;Kmd6bfl/fl embryos had a defect in osteogenic differentiation by analysis of Runx2 and Sp7 expression and by using an in vitro osteogenic differentiation assay. The defect in palatogenesis in the Wnt1-Cre;Kmd6bfl/fl embryos correlated with increased levels of H3K27me3 in the palatal mesenchyme cells. By genetically reducing the dosage of Ezh2, which is a major H3K27 methyltransferase, they show that Ezh2 heterozygosity reduced the H3K27me3 in the palatal mesenchyme and partly rescued cleft palate in the Wnt1-Cre;Kmd6bfl/fl;Ezh2+/- embryos, indicating that antagonistic actions of Ezh2 and Kdm6b in H3K27 methylation are important in regulating palatogenesis. They performed RNA-seq analysis and found that the Wnt1-Cre;Kmd6bfl/fl embryos exhibited significant down-regulation of expression of 382 genes and significant upregulation of 259 genes in the developing palatal shelves at E12.5. Ingenuity pathway analysis of the RNA-seq data showed that the "Role of BRCA1 in DNA Damage" pathway was most significantly upregulated, whereas Gene Ontology analysis showed that the "p53 pathway" was among the top 10 pathways affected. They validated that the Wnt1-Cre;Kmd6bfl/fl embryos had increased DNA damage (marked by H2AX foci) in proliferating cells in the palatal mesenchyme and reduced expression of p53 mRNAs. Remarkably, treating the Wnt1-Cre;Kmd6bfl/fl embryos in utero through maternal IP injection of Nutlin-3, a small molecule inhibitor of MDM2-p53 interactions that causes stabilization of p53 protein, was able to rescue the cleft palate defect, indicating that repression of p53 expression in the Wnt1-Cre;Kmd6bfl/fl embryos plays an important role in the cleft palate pathogenesis. The manuscript further shows that the Wnt1-Cre;Kmd6bfl/fl embryonic palate had increased H3K27me3 at the p53 gene promoter region by ChIP-qPCR analysis. They performed ATAC-seq analysis of E13.5 wildtype embryonic palate and found that an ATAC-seq peak in the p53 gene promoter region contained E2F transcription factor binding motif. They show that Tfdp1, a member of the E2F family, was expressed in the embryonic palate and its expression was unaltered in the Wnt1-Cre;Kmd6bfl/fl embryonic palate. The performed ChIP-qPCR, of which the result suggested decreased Tfdp1 occupancy at the p53 gene promoter in the Wnt1-Cre;Kmd6bfl/fl embryonic palate than in control samples. They show that siRNA knockdown of Tfdp1 in primary palatal mesenchyme cells resulted in decrease in p53 mRNA expression and that overexpression of exogenous Tfdp1 caused increased p53 mRNA expression in control but not the Wnt1-Cre;Kmd6bfl/fl embryonic palate cells. In addition, they performed co-immunoprecipitation analysis and found that Tfdp1 and Kdm6b proteins were pulled down together from embryonic palatal extracts. They concluded that these results collectively suggested that Kdm6b and Tfdp1 work together to activate p53 expression in the palatal mesenchyme and play an important role in regulating palatogenesis.

In addition, this study generated RNA-seq data from E12.5 control and the Wnt1-Cre;Kmd6bfl/fl mutant embryonic palatal shelves, as well as ATAC-seq data from E13.5 wildtype mouse embryonic palatal tissues, which will serve as useful resources for palate development research. However, as written, the results do not directly address the stated major aim for investigating "how epigenetic regulators coordinate with tissue-specific regulatory factors during morphogenesis of specific organs".

The title, " Kdm6b confers Tfdp1 with the competence to activate p53 signaling in regulating palatogenesis", is not an accurate summary of the findings presented. First, the manuscript reports that Kdm6b mutant mice exhibit 90% penetrance of cleft palate, but among thousands of p53-null mice that have been studied in the last two decades few p53-null mice have been shown to have a cleft palate (of many primary research papers on studies of p53-null mice, only Tateossian et al. 2015 reported cleft palate in 2 out of 10 p53 homozygous mutant mice in one particular genetic background). Thus, the decreased expression of p53 in the palatal mesenchyme alone could not account for the cause of the disruptions in palate development in the Kdm6b mutant mice even though Nutlin-3 treatment, presumably through stabilization of the p53 protein, was able to partly rescue palate morphogenesis in the Kdm6b mutant embryos. Second, no data has been shown to indicate that Tfdp1 mediated activation of p53 is required for palate development. Third, the data presented indicate that Kdm6b function in the neural crest cells resulted in reduction in the H3K27me3 repression mark in the palatal mesenchyme cells, including in the p53 gene promoter region near where Tfdp1 binds, but there is no evidence that Kdm6b function is specific for Tfdp1 binding nor whether Kdm6-mediated demethylation of H3K27me3 at the p53 gene region is mediated by interaction with Tfdp1. Tfdp1 could be one of many transcription factors that could only bind to their target sites after heterochromatin-like repressive marks are removed.

Many conclusions and/or interpretation of the results, particularly the statements in the Abstract, in the manuscript were inaccurate.

1. The third and fourth sentences (Lines 36 – 37) in the Abstract appear to suggest that the major aim of the study was to use palatogenesis as a model to answer the question "how epigenetic regulators coordinate with tissue-specific regulatory factors during morphogenesis of specific organs", but the study does not address this aim at all. No "coordination" between Kdm6b mediated epigenetic regulation with any "tissue-specific" regulatory factor is demonstrated. The data show that lack of Kdm6b-mediated H3K27me3 demethylation in the neural crest cells resulted in repression of many genes, including p53, in the developing palatal mesenchyme and that Tfdp1 binding and activation of the p53 gene promoter may require prior Kdm6b-mediated removal of the H3K27me3 repressive mark.

2. Most of the conclusions/statements written in the Abstract were not accurate or not sufficiently supported by the data:

2a. Lines 41 – 42 (and similar sentences in Lines 99 – 100, Line 310, ) state, "activity of H3K27me3 on the promoter of p53 is precisely controlled by Kdm6b, and Ezh2 in regulating p53 expression in cranial neural crest cells". The data presented indicate opposing effects of Kdm6b and Ezh2 on the total amount of H3K27me3 in the palatal tissues and on the levels of p53 mRNA, but does not address how "precisely" they control of H3K27me3 levels at any particular locus.

2b. Lines 42 – 44 state, "Kdm6b renders chromatin accessible to the transcription factor Tfdp1, which binds to the promoter of p53 along with Kdm6b to specifically activate p53 expression during palatogenesis". There is no data presented to support for the second half of the sentence (and similar statements in Lines 101 and 426). There is no data showing binding of Tfdp1 and Kdm6b together at the p53 gene promoter. There is no data showing that binding of Tfdp1 at the p53 promoter is required for the p53 gene expression in the palatal mesenchyme. There is no data showing Kdm6b activates p53 expression through direct interaction with Tfdp1. There is no data showing that Tfdp1 mediated activation of p53 specifically occurs in the palatal tissues but not in other tissues when there is cellular stress.

2c. Lines 44 – 46, "our results highlight the important role of the epigenetic regulator Kdm6b and how it cooperates with Tfdp1 to achieve its functional specificity in regulating p53 expression,…" (and similar sentence in Lines 494 – 495). While the data showed that overexpression of Tfdp1 resulted in increased p53 mRNA expression in cultured palatal mesenchyme cells from control but not Kdm6b mutant embryos and the decrease in p53 mRNAs in the Kdm6b mutant palatal mesenchyme correlated with reduced Tfdp1 binding at the p53 gene promoter, the study has not provided evidence for direct cooperation between Kdm6b and Tfdp1 at the p53 gene promoter. It is quite possible that another unidentified factor interacts with Kdm6b to target Kdm6b to the p53 gene region among many other chromatin regions to remove the H3K27me3 repression mark in the developing palatal mesenchyme and Tfdp1 is only able to bind to the p53 gene promoter after the K3K27me3 demethylation. The results, particularly the demonstration of Ntlin-3 mediated rescue of palatal morphogenesis in the Kdm6b mutants, may have broad implications beyond Kdm6b mutant mouse model such that pharmacologically induced stabilization of p53 may be applicable for therapeutic intervention in cases where genetic-environment interactions disrupt developmental or other cellular processes while also inhibiting stress-induced activation of p53.

3. Lines 133 – 135 state, " although the palatal shelves were able to elevate, the maxilla and palatine bones, as well as the palate stromal mesenchyme and soft palate muscles, failed to grow towards the midline in Wnt1-Cre;Kdm6bfl/fl mice". However, the histology data in Figure 2—figure supplement 4, panels C/D/G/H, clearly show a failure or delay in palatal shelf elevation at E14.0 and E14.5 in the mutant embryos. The manuscript needs to accurately describe the results and investigate whether delay in palatal shelf elevation was the likely cause of cleft palate in the mutant.

4. Some numbers used to describe the results were inaccurate. For examples, Lines 119 – 120, "Loss of Kdm6b in CNC-derived cells resulted in complete cleft palate in Wnt1-Cre;Kmd6bfl/fl mice (90% phenotype penetrance, N=7)."

5. Given that the manuscript is focused on the effect of Kdm6b on the regulation of p53 in palatogenesis, the authors should discuss why few p53-/- mice had cleft palate, but Nutlin3 treatment was able to rescue the cleft palate defect in Kdm6b mutant mice at high efficiency. The manuscript indicated N=5 for the Nutlin-3 rescue experiment but did not indicate whether all 5 mutants were rescued. In addition, whereas the p53 mRNAs were reduced by about 50% in the Kdm6b mutant palate, the western blot in Figure 4—figure supplement 5F appears to show that Nutlin-3 treatment resulted in similar or more p53 protein in the mutant sample than the control. It would be helpful to quantify the relative levels of p53 protein in the Nutlin-3 treated mutant and control samples, and to discuss, if that is the result, how Nutlin-3 could induce p53 protein more efficiently in the mutant than in the control samples.

*Reviewer #2 (Recommendations for the authors):*

I have just a few comments that would not change the interpretation of the data nor would I request additional experiments.

1. Figure 2 - Figure Supplement 4, (A-B). Wnt1-Cre;Kdm6bfl/fl mice are compared to Wnt1-Cre;Kdm6bfl/+ , but why aren't they compared to a wild type control instead?

2. Figure 2 - Figure Supplement 4, (E-H). The TUNEL stain data should probably be quantified by using TUNEL+ cell numbers collected across several different sections/samples. The difference between the control and the mutant, as seen in the 4 images provided, is large if expressed as a percentage but still limited to 3 cells or fewer (especially at day E13.5). Even at E14.5, while there are 3 TUNEL+ cells seen in the palatal shelves of the mutant and 0 in the control, multiple TUNEL signals can be seen just outside of the dotted region in the control but not in the mutant; either the control has more cell death right next to the palatal shelves, or the small difference between the two can be explained as a sampling error without more data points.

3. L181-185, figure 2I-P; Similarly, can the authors state the sample size and quantify the RUNX2+ and SP7+ cells for easier comparison?

4. L213-219, figure 3B-C; Can the authors elaborate more on the identification of p53 from the analysis of the RNA-seq? As "epithelial adherens junction signaling" in IPA also corroborate with "cadherin signaling pathway" in GO. Same for "regulation of Actin-based motility by Rho" in IPA and "cytoskeletal regulation by Rho GTPase" in GO. Is there any other highly confident targets that are worth exploring?

5. Figure 4 - Figure Supplement 5, (F). The p53 band intensity difference between the control and the mutant in the 10% DMSO group is slight; the loading control band (actin) also demonstrates a small difference in size that follows the same pattern. It would be better to see the p53 band difference quantified.

6. L323-334, figure 5F-N; In contrast to what the authors claim, Ezh1 seems to have a stronger signal than Ezh2. Besides, the expression pattern between Ezh1 and Ezh2 is quite different. Can the authors support the claim by using normalized quantification? Otherwise, can the authors elaborate why they chose to KO Ezh2 in the Wnt1-Cre; Kdm6bfl/fl model?

7. Paper shows that inhibition of p53 by siRNA results in increased CNCC proliferation and DNA damage and reduces markers of osteogenesis, as seen in the siRNA treated cells. But is it possible to treat an embryo with a p53 inhibitor to directly demonstrate that loss of p53 in this model produces cleft palate? For example, Jones et al. treated pregnant mice with pifithrin-α to inhibit p53 (Nat. Med. 2011).

8. The paper describes how the control cells were treated with Tfdp1 siRNA, which resulted in lower levels of p53. Why wasn't the same treatment applied to the Wnt1-Cre;Kdm6bfl/fl cells, especially if both the controls and the mutants were later treated with the Tfdp1 overexpression plasmid?

*Reviewer #3 (Recommendations for the authors):*

In the present study, Tingwei Guo et al. use the mouse secondary palate as a model to assess the function of Kdm6b, a H3K27me3 demethylase, in the regulation of embryonic development. Guo's study shows that Kdm6b plays an essential role in neural crest development, and that loss of Kdm6b perturbs p53 pathway-mediated activity, leading to complete clefting of the secondary palate along with cell proliferation and differentiation defects.

In addition, the study reveals that Kdm6b and Ezh2 control p53 expression in cranial neural crest cells and that Kdm6b renders chromatin accessible to the transcription factor TFDP1 to activate p53 expression during palatogenesis. Together, the findings presented in this manuscript highlight the important role of the epigenetic regulator KDM6B and how it cooperates with TFDP1 to achieve its functional specificity in controlling p53 expression, and further provide mechanistic insights into the epigenetic regulatory network during secondary palate organogenesis.

– Over the last years, it has been reported by multiple groups that among the various layers of epigenetic regulation, DNA methylation and histone methylation are key drivers of diverse cellular events and developmental processes. In addition, it has been demonstrated that demethylation also plays important roles during development. For instance, demethylation of H3K4 is required for maintaining pluripotency in embryonic stem cells, and the demethylases KDM6A and KDM6B are required for proper gene expression. Indeed, the concept that failure to maintain epigenomic integrity can cause deleterious consequences for embryonic development has been extensively explored by various groups and is not novel per se. In addition, both lysine methyltransferase Kmt2a and demethylase Kdm6a have been recently shown to be essential for cardiac and neural crest development. For example, Shpargel reported that mice carrying neural crest deletion of Kdm6a exhibit craniofacial defects, including cleft or arched palate, cardiac abnormalities, and postnatal growth retardation, modeling the clinical features of Kabuki syndrome (Shpargel et al. PNAS, 2017). In summary, roles of these demethylases in neural crest development are already known. However, how these epigenetic changes lead to tissue-specific responses during neural crest fate determination and differentiation remains poorly understood and understudied, making the current manuscript of interest and timely.

– Epigenetic regulation plays extensive roles in development and diseases and its disruption not only can cause multiple developmental defects, but also increases the risk of neoplastic transformation. However, our knowledge of how epigenetic regulators coordinate with tissue-specific regulatory factors to control tissue and organ morphogenesis is still rudimentary. Therefore, the present paper will be of interest to the craniofacial biology community and to the broader developmental biology community, as well as to all those devoted to the study of the epigenetic and transcriptional regulation of morphogenesis and organogenesis.

– The study is robust, detailed, and comprises a wealth of original results and data of high quality, illustrated through many elegant figures. There are only some points of concern that need to be addressed, mainly related to additional quantitative analyses that are required for some of the experiments discussed in the manuscript and the need for clarifications regarding the regulation of the p53 pathway.

– Nomenclature: protein names should be written in uppercase throughout the text and in the figures. In the current manuscript use of the nomenclature is not consistent. Often protein and gene names are not listed correctly. See: http://www.informatics.jax.org/mgihome/nomen/gene.shtml#ps

– Figure 1: In Wnt1-Cre;Kdm6bfl/fl embryos the tongue appears to be grossly dysmorphic and abnormally positioned (e.g. see panel Q). If this is only due to a technical artifact of the section, then a better image should be chosen for Figure 1Q. If the tongue is instead abnormally formed and enlarged, then this result should be discussed and well documented, given that withdrawal of the tongue from between the vertical palatal shelves is required for their seamless fusion and closure during embryonic development.

– Figure 2 – Figure Suppl 4E-H: The Authors state that there is increased cell death in the palatal mesenchyme of Wnt1-Cre;Kdm6bfl/fl mice compared to controls, based on TUNEL staining, as shown in the figure. It is difficult to be convinced of this result, given that the TUNEL-positive cells are extremely sparse in both the control and mutant shown in these panels. If the Authors are convinced of their finding based on multiple experiments, they should show sections that better illustrate the defect, as well as quantify the numbers of TUNEL-positive cells over multiple sections for both control and mutant samples. In addition, they should include a graph comprising the total numbers of cells that were counted, the percentage of TUNEL-positive cells in both control and mutant, the fold increase of TUNEL-positive cells in the mutant, and the statistical significance. Unless these quantitative experiments are included, the Authors should definitely delete "increased cell death" for the mutant from the summary model shown in Figure 8.

However, if the Authors were to prove the presence of an increase of cell death in the palatal mesenchyme of Wnt1-Cre;Kdm6bfl/fl mice by conducting additional experiments, this finding would somehow contradict the result – which is very convincing – that p53 expression is decreased in mutant tissue (shown currently in Figure 7). Interestingly, in the discussion the authors mention (line 517): "In this study we notice that downregulated expression of p53 in Wnt1-Cre;Kdm6bfl/fl mice results in hyperproliferation and increased DNA damage in the proliferative cells, which might further lead to cell senescence." Assessing cell senescence, instead of cell death, in the mutant compared to the control could be revealing.

– Figure 2I-L: The Authors state: "There was a decrease in the number of Runx2+ cells in the palatal mesenchyme at both E13.5 and E14.5 in Wnt1-184 Cre;Kdm6bfl/fl mice in comparison to the control" (Line 182-184). Either the Authors quantify the "number of cells" (see also comment above), or they could do a qRT-PCR to evaluate Runx2 mRNA in mutant versus control tissues.

– Figure 3B: It would be beneficial to separate the genes that are upregulated from the genes that are downregulated in Wnt1-Cre;Kdm6bfl/fl mice versus controls. The overall message would be much clearer.

– Figure 6: The rescue experiments are one of the main strengths of this study: beautiful and rigorous use of genetics to validate the pathways under analysis! However, it would add a great deal of strength to the rescue experiments to also evaluate the rescue at the cellular level (i.e. assess the rescue of proliferation and/or DNA damage, even if partial, or alternatively, examine whether expression of Sp7 or Runx2 is rescued by qRT-PCR).

– Figure 7, panel I and Figure 7 – Figure Suppl 6: Red box that is described as highlighting the p53 promoter is placed instead over the promoter of a different gene, Wrap53. Indeed, this red box contains the TSS of Wrap53 and not p53. The Wrap53 gene encodes an essential component of the telomerase holoenzyme complex, a ribonucleoprotein complex required for telomere synthesis. The encoded protein interacts with other components of active telomerase and with small Cajal body RNAs (scaRNAs), which are involved in modifying splicing RNAs. It was reported that Wrap53 also functions as a p53 antisense transcript, which regulates endogenous p53 mRNA levels and the levels of P53 protein by targeting the 5' untranslated region of p53 mRNA (Mahmoudi et al. Mol Cell 2009). Therefore, Wrap53 can indirectly alter p53 levels of expression and regulation. siRNA knockdown of Wrap53 results in significant decrease in p53 mRNA and suppression of P53 induction upon DNA damage. Conversely, overexpression of Wrap53 increases p53 mRNA and protein levels (Mahmoudi et al. Molecular Cell 2009; Farnebo et al., Cell Cycle 2009; Saldana-Meyer et al., Genes and Development 2013). These studies unequivocally demonstrated that Wrap53 regulates p53. As it turns out, the primers used for the ChIP-qPCR experiment shown in Figure 7 were now blasted and they correspond to the Wrap53 promoter and not to the p53 promoter. Given all the knowledge discussed above, the current statement: "factor Tfdp1 binds to the promoter of p53 along with Kdm6b to specifically activate the expression of this tumor suppressor gene" should be re-evaluated and clarified. This point should be better investigated. As the distance between the Wrap53 and p53 genes is only approximately 1kb, in additional experiments the Authors should use primers specific to the p53 promoter and to the Wrap53 promoter together with chromatin sonicated to an average size of 300-600 bp, which should provide a clear answer as to whether TFDP1 binds directly to the p53 promoter, or to the Wrap53 promoter, or to both. If TFDP1 does not bind directly to p53 but to the Wrap53 promoter instead, regulation of p53 transcription would then be indirect, via Wrap53. This finding would still be of interest. This reviewer believes that this point is important and should be adequately clarified and examined in further depth.

– Figure 7 – Figure Suppl 6: The Authors also analyze expression patterns of Kdm6b and its co-expression with Tfdp1 by scRNAseq. The scRNAseq data sets have already been described by the same Authors in a(nother) interesting paper that was recently published. There appears to be a striking enrichment of both Kdm6b and Tfdp1 genes in a particular cell subpopulation emerging from the scRNAseq datasets. It would be very interesting to know which cell type comprises this particular cluster? Also, the individual Panels in B should be enlarged to better visualize each cell clusters. It would also be very helpful to the reader to list the specific cell types that comprise each cluster in the Panels in B.

---

## [Author Response]

Essential revisions:The reviewers agree that this is an interesting paper with the wealth of high quality data/ However, there are several areas that need further experimentation and clarification.1) The results as written do not fully address the stated major aim of the study, i.e. investigating "how epigenetic regulators coordinate with tissue-specific regulatory factors during morphogenesis of specific organs". This point is overemphasized in the manuscript but not fully developed so should be toned down. As case in point, the abstract overstates and goes well beyond the data presented.

We thank reviewers for this comment, and we have added two experiments addressing the reviewer’s concern. One is ChIP-qPCR, which showed KDM6B is deposited at the promoter region of *Trp53*. Another experiment we have added used *Kdm6b-* and *Kdm6a*-overexpressing (OE) plasmids to transfect cells from the palatal mesenchyme of *Kdm6b* mutant mice. Expression of *Trp53* increased in the cells transfected with *Kdm6b* OE plasmid, but not in the cells transfected with *Kdm6a* OE plasmid. These results suggested that not all the histone demethylases can activate expression of *Trp53* in the palatal mesenchyme and that *Kdm6b* is the critical and functional specific for activating expression of *Trp53* during palatogenesis. We also adjusted our statement in the manuscript, which has been highlighted in the revised version.

2) The regulations of the p53 pathway requires additional and deeper investigations plus added discussion.

We thank the reviewers for this suggestion. As we mentioned above, we have added two experiments regarding regulation of *Trp53*. We have also added some discussion regarding the phenotypes of *Trp53^-/-^* mice per the reviewers’ suggestions, which is highlighted in the Discussion section.

3) Additional quantitative analyses are required to support claims regarding increased numbers of apoptotic cells and decreased numbers of Runx2 positive cells in mutant palate mesenchyme.

We have assessed cellular senescence instead of cell death according to the reviewer’s suggestion. Data has been added to Figure2—figure supplement 1E-G. Both numbers of RUNX2+ and SP7+ cells in Figure 2 have been quantified per the reviewers’ suggestion.

Reviewer #1 (Recommendations for the authors):[…]Many conclusions and/or interpretation of the results, particularly the statements in the Abstract, in the manuscript were inaccurate.1. The third and fourth sentences (Lines 36 – 37) in the Abstract appear to suggest that the major aim of the study was to use palatogenesis as a model to answer the question "how epigenetic regulators coordinate with tissue-specific regulatory factors during morphogenesis of specific organs", but the study does not address this aim at all. No "coordination" between Kdm6b mediated epigenetic regulation with any "tissue-specific" regulatory factor is demonstrated. The data show that lack of Kdm6b-mediated H3K27me3 demethylation in the neural crest cells resulted in repression of many genes, including p53, in the developing palatal mesenchyme and that Tfdp1 binding and activation of the p53 gene promoter may require prior Kdm6b-mediated removal of the H3K27me3 repressive mark.

We thank the reviewer for this comment and have rewritten these sentences as follows: “However, the question of how epigenetic changes lead to tissue-specific responses during neural crest fate determination and differentiation remains understudied.”

2. Most of the conclusions/statements written in the Abstract were not accurate or not sufficiently supported by the data:2a. Lines 41 – 42 (and similar sentences in Lines 99 – 100, Line 310, ) state, "activity of H3K27me3 on the promoter of p53 is precisely controlled by Kdm6b, and Ezh2 in regulating p53 expression in cranial neural crest cells". The data presented indicate opposing effects of Kdm6b and Ezh2 on the total amount of H3K27me3 in the palatal tissues and on the levels of p53 mRNA, but does not address how "precisely" they control of H3K27me3 levels at any particular locus.

We thank the reviewer for this comment and have rewritten these sentences as follows: “Furthermore, activity of H3K27me3 on the promoter of *Trp53* is antagonistically controlled by *Kdm6b* and *Ezh2* in cranial neural crest cells.”

2b. Lines 42 – 44 state, "Kdm6b renders chromatin accessible to the transcription factor Tfdp1, which binds to the promoter of p53 along with Kdm6b to specifically activate p53 expression during palatogenesis". There is no data presented to support for the second half of the sentence (and similar statements in Lines 101 and 426). There is no data showing binding of Tfdp1 and Kdm6b together at the p53 gene promoter. There is no data showing that binding of Tfdp1 at the p53 promoter is required for the p53 gene expression in the palatal mesenchyme. There is no data showing Kdm6b activates p53 expression through direct interaction with Tfdp1. There is no data showing that Tfdp1 mediated activation of p53 specifically occurs in the palatal tissues but not in other tissues when there is cellular stress.

We thank the reviewer for this comment and have rewritten these sentences as follows: “More importantly, without *Kdm6b*, the transcription factor TFDP1, which normally binds to the promoter of *Trp53*, cannot activate *Trp53* expression in palatal mesenchymal cells. Furthermore, the function of *Kdm6b* in activating *Trp53* in these cells cannot be compensated for by the closely related histone demethylase *Kdm6a*.”

2c. Lines 44 – 46, "our results highlight the important role of the epigenetic regulator Kdm6b and how it cooperates with Tfdp1 to achieve its functional specificity in regulating p53 expression,…" (and similar sentence in Lines 494 – 495). While the data showed that overexpression of Tfdp1 resulted in increased p53 mRNA expression in cultured palatal mesenchyme cells from control but not Kdm6b mutant embryos and the decrease in p53 mRNAs in the Kdm6b mutant palatal mesenchyme correlated with reduced Tfdp1 binding at the p53 gene promoter, the study has not provided evidence for direct cooperation between Kdm6b and Tfdp1 at the p53 gene promoter. It is quite possible that another unidentified factor interacts with Kdm6b to target Kdm6b to the p53 gene region among many other chromatin regions to remove the H3K27me3 repression mark in the developing palatal mesenchyme and Tfdp1 is only able to bind to the p53 gene promoter after the K3K27me3 demethylation. The results, particularly the demonstration of Ntlin-3 mediated rescue of palatal morphogenesis in the Kdm6b mutants, may have broad implications beyond Kdm6b mutant mouse model such that pharmacologically induced stabilization of p53 may be applicable for therapeutic intervention in cases where genetic-environment interactions disrupt developmental or other cellular processes while also inhibiting stress-induced activation of p53.

We thank the reviewer for this comment and have rewritten these sentences as follows: “Collectively, our results highlight the important role of the epigenetic regulator KDM6B and how it specifically interacts with TFDP1 to achieve its functional specificity in regulating *Trp53* expression, and further provide mechanistic insights into the epigenetic regulatory network during organogenesis.”

3. Lines 133 – 135 state, " although the palatal shelves were able to elevate, the maxilla and palatine bones, as well as the palate stromal mesenchyme and soft palate muscles, failed to grow towards the midline in Wnt1-Cre;Kdm6bfl/fl mice". However, the histology data in Figure 2—figure supplement 4, panels C/D/G/H, clearly show a failure or delay in palatal shelf elevation at E14.0 and E14.5 in the mutant embryos. The manuscript needs to accurately describe the results and investigate whether delay in palatal shelf elevation was the likely cause of cleft palate in the mutant.

We thank the reviewer for this comment. In *Wnt1^Cre^;Kdm6b^fl/fl^* mice, the development of the palate is generally delayed compared to the control mice. There is some variation at E14.5, but in most of the samples we investigated, the palatal shelves were able to elevate in *Kdm6b* mutant mice, as shown in Figure2 L and N. All the samples at E15.5 are accurately represented in Figure2 P. Furthermore, all the *Kdm6b* mutant mice are accurately represented by Figure1 M-R at newborn stage. We have never observed palatal shelves oriented vertically beside the tongue from E15.5 to newborn stage (n>10). Therefore, we conclude there is no elevation defect in *Kdm6b* mutant mice.

4. Some numbers used to describe the results were inaccurate. For examples, Lines 119 – 120, "Loss of Kdm6b in CNC-derived cells resulted in complete cleft palate in Wnt1-Cre;Kmd6bfl/fl mice (90% phenotype penetrance, N=7)."

We thank reviewer for this comment and have edited accordingly.

5. Given that the manuscript is focused on the effect of Kdm6b on the regulation of p53 in palatogenesis, the authors should discuss why few p53-/- mice had cleft palate, but Nutlin3 treatment was able to rescue the cleft palate defect in Kdm6b mutant mice at high efficiency. The manuscript indicated N=5 for the Nutlin-3 rescue experiment but did not indicate whether all 5 mutants were rescued. In addition, whereas the p53 mRNAs were reduced by about 50% in the Kdm6b mutant palate, the western blot in Figure 4—figure supplement 5F appears to show that Nutlin-3 treatment resulted in similar or more p53 protein in the mutant sample than the control. It would be helpful to quantify the relative levels of p53 protein in the Nutlin-3 treated mutant and control samples, and to discuss, if that is the result, how Nutlin-3 could induce p53 protein more efficiently in the mutant than in the control samples.

We thank the reviewer for this comment regarding *Trp53^-/-^* mice, which show a low penetrance of cleft palate. Regarding this question, we think *Kdm6b* as an epigenetic regulator acts on more downstream targets than just *Trp53.* In *Trp53^-/-^* mice, there might be other factors that could compensate for the lost function of *Trp53*. However, in *Kdm6b* mutant mice, this compensation cannot occur. We have also added this point in our Discussion section. Nutlin-3 treatment successfully rescued the hard palate cleft in all observed *Wnt1^Cre^;Kdm6b^fl/fl^* mice (N = 5). Two of these five showed a posterior soft palate cleft. We appreciate the reviewer’s comments on this detail and its improvement to the discussion in our manuscript.

We have added quantification for the Western blot results in Figure 4—figure supplement 1G as per the reviewer’s suggestion. Regarding why Nutlin-3 is more efficient in the mutant than in the controls, we cannot offer a specific explanation but note that this phenomenon is very common with Nutlin-3 treatment. For example, in Li et al.’s study, they also observed Nutlin-3 treatment is more efficient in knockout cells compared to controls (Li et al. 2016).

Reviewer #2 (Recommendations for the authors):I have just a few comments that would not change the interpretation of the data nor would I request additional experiments.1. Figure 2 - Figure Supplement 4, (A-B). Wnt1-Cre;Kdm6bfl/fl mice are compared to Wnt1-Cre;Kdm6bfl/+ , but why aren't they compared to a wild type control instead?

We thank the reviewer for this comment. We compared *Wnt1^Cre^;Kdm6b^fl/+^;Rosa26-CAG-tdTomato* to a wild type control and no differences were observed between these two. When we collected samples for this experiment, we compared the *Kdm6b* mutant mice to littermate controls to avoid developmental variation between different litters. Ratio to get *Wnt1^Cre^;Kdm6b^fl/+^* is higher than wild type. The picture we used in the figure is of *Wnt1^Cre^;Kdm6b^fl/+^;Rosa26-CAG-tdTomato.*

2. Figure 2 - Figure Supplement 4, (E-H). The TUNEL stain data should probably be quantified by using TUNEL+ cell numbers collected across several different sections/samples. The difference between the control and the mutant, as seen in the 4 images provided, is large if expressed as a percentage but still limited to 3 cells or fewer (especially at day E13.5). Even at E14.5, while there are 3 TUNEL+ cells seen in the palatal shelves of the mutant and 0 in the control, multiple TUNEL signals can be seen just outside of the dotted region in the control but not in the mutant; either the control has more cell death right next to the palatal shelves, or the small difference between the two can be explained as a sampling error without more data points.

We appreciate the reviewer’s comment. A similar comment was also brought up by another reviewer. We have assessed cell senescence instead of cell death using primary palatal cell according to another reviewer’s suggestion. The data has been added to Figure 2-figure supplement.

3. L181-185, figure 2I-P; Similarly, can the authors state the sample size and quantify the RUNX2+ and SP7+ cells for easier comparison?

We thank the reviewer for this comment. Both RUNX2+ and SP7+ cells have been quantified. The results have been added to Figure 2.

4. L213-219, figure 3B-C; Can the authors elaborate more on the identification of p53 from the analysis of the RNA-seq? As "epithelial adherens junction signaling" in IPA also corroborate with "cadherin signaling pathway" in GO. Same for "regulation of Actin-based motility by Rho" in IPA and "cytoskeletal regulation by Rho GTPase" in GO. Is there any other highly confident targets that are worth exploring?

We appreciate the reviewer’s comment. In GO we did notice Wnt signaling pathway, which also has some crosstalk with Cadherin signaling, ranking as the top signaling pathway. We assessed expression of Axin2 in our sample using in situ RNAscope hybridization, but didn’t observe a difference between control and *Kdm6b* mutant mice. We agree with the reviewer that there are more signaling pathways that are worthy to investigate, but are beyond the scope of this study.

5. Figure 4 - Figure Supplement 5, (F). The p53 band intensity difference between the control and the mutant in the 10% DMSO group is slight; the loading control band (actin) also demonstrates a small difference in size that follows the same pattern. It would be better to see the p53 band difference quantified.

We thank the reviewer for this comment. We added a quantified result using Image J based on the integrated density of the bands of 3 individual samples. The data has been added to Figure 4-figure supplement 1G.

6. L323-334, figure 5F-N; In contrast to what the authors claim, Ezh1 seems to have a stronger signal than Ezh2. Besides, the expression pattern between Ezh1 and Ezh2 is quite different. Can the authors support the claim by using normalized quantification? Otherwise, can the authors elaborate why they chose to KO Ezh2 in the Wnt1-Cre; Kdm6bfl/fl model?

We appreciate the reviewer’s comment regarding *Ezh1*. Our in vivo staining of *Ezh1* was done using RNAscope, while that of EZH2 was immunostaining. It’s hard to quantitatively compare these two signals directly based on the staining. However, we agree with the reviewer that the expression patterns of *Ezh1* and *Ezh2* are different. Expression of *Ezh1* is more enriched in the oral side of the palatal shelf, while expression of *Ezh2* is more universal. However, from the Western blot we could see at the protein level, the expression of EZH2 is higher than EZH1 when we load the same amount of total protein (based on a beta-actin internal control). Furthermore, conventional knockout *Ezh1* mice don’t have any abnormal phenotypes (http://www.informatics.jax.org/marker/MGI:1097695), while *Wn1^Cre^;Ezh2^fl/fl^* mice show severe craniofacial defects. These results suggest that *Ezh2* may have a more important role in regulating CNCCs.

7. Paper shows that inhibition of p53 by siRNA results in increased CNCC proliferation and DNA damage and reduces markers of osteogenesis, as seen in the siRNA treated cells. But is it possible to treat an embryo with a p53 inhibitor to directly demonstrate that loss of p53 in this model produces cleft palate? For example, Jones et al. treated pregnant mice with pifithrin-α to inhibit p53 (Nat. Med. 2011).

We thank the reviewer for this comment. We did generate *Wnt1^Cre^;Trp53^fl/fl^* mice, and only about 20% of the mutant mice showed a cleft palate phenotype. This ratio is similar to what has been previously reported for *Trp53^-/-^* mice. Regarding this question, we think *Kdm6b* as an epigenetic regulator affects more downstream targets than just *Trp53.* In *Trp53^-/-^* mice, there might be other factors that could compensate for the lost function of *Trp53*. However, in *Kdm6b* mutant mice, this compensation cannot occur. We have also added this point to our Discussion section.

8. The paper describes how the control cells were treated with Tfdp1 siRNA, which resulted in lower levels of p53. Why wasn't the same treatment applied to the Wnt1-Cre;Kdm6bfl/fl cells, especially if both the controls and the mutants were later treated with the Tfdp1 overexpression plasmid?

We thank the reviewer for this comment. The purpose of treating control cells with *Tfdp1* siRNA is to test whether expression of *Trp53* is regulated by the transcription factor TFDP1, which is validated by ATAC-seq and ChIP-qPCR results. Therefore, we used cells from the palatal mesenchyme of control mice to test the function of TFDP1 in regulating *Trp53*. Expression of *Trp53* is already reduced in the *Kdm6b* mutant mice, so we think using cells from control mice is more appropriate for this experimental purpose.

Reviewer #3 (Recommendations for the authors):[…]– Nomenclature: protein names should be written in uppercase throughout the text and in the figures. In the current manuscript use of the nomenclature is not consistent. Often protein and gene names are not listed correctly. See: http://www.informatics.jax.org/mgihome/nomen/gene.shtml#ps

We appreciate the reviewers’ comments on this issue. All the protein and gene names have been corrected in the text and figures.

– Figure 1: In Wnt1-Cre;Kdm6bfl/fl embryos the tongue appears to be grossly dysmorphic and abnormally positioned (e.g. see panel Q). If this is only due to a technical artifact of the section, then a better image should be chosen for Figure 1Q. If the tongue is instead abnormally formed and enlarged, then this result should be discussed and well documented, given that withdrawal of the tongue from between the vertical palatal shelves is required for their seamless fusion and closure during embryonic development.

We really appreciate the reviewer’s comment and carefully went through our samples to examine the tongue morphology, including CT images which provide a better view of the size of the tongue (Figure 1 E-F are representative CT images). We didn’t observe any obvious size differences between control and mutant mice. Better images have been used for Figure 1Q.

– Figure 2 – Figure Suppl 4E-H: The Authors state that there is increased cell death in the palatal mesenchyme of Wnt1-Cre;Kdm6bfl/fl mice compared to controls, based on TUNEL staining, as shown in the figure. It is difficult to be convinced of this result, given that the TUNEL-positive cells are extremely sparse in both the control and mutant shown in these panels. If the Authors are convinced of their finding based on multiple experiments, they should show sections that better illustrate the defect, as well as quantify the numbers of TUNEL-positive cells over multiple sections for both control and mutant samples. In addition, they should include a graph comprising the total numbers of cells that were counted, the percentage of TUNEL-positive cells in both control and mutant, the fold increase of TUNEL-positive cells in the mutant, and the statistical significance. Unless these quantitative experiments are included, the Authors should definitely delete "increased cell death" for the mutant from the summary model shown in Figure 8.However, if the Authors were to prove the presence of an increase of cell death in the palatal mesenchyme of Wnt1-Cre;Kdm6bfl/fl mice by conducting additional experiments, this finding would somehow contradict the result – which is very convincing – that p53 expression is decreased in mutant tissue (shown currently in Figure 7). Interestingly, in the discussion the authors mention (line 517): "In this study we notice that downregulated expression of p53 in Wnt1-Cre;Kdm6bfl/fl mice results in hyperproliferation and increased DNA damage in the proliferative cells, which might further lead to cell senescence." Assessing cell senescence, instead of cell death, in the mutant compared to the control could be revealing.

We really appreciate this suggestion. We have assessed cell senescence instead of cell death using primary palatal cell culture. Cells from *Wnt1^Cre^;Kdm6b^fl/fl^* mice showed increased cellular senescence compared to the cells from control mice. The data has been added to the Figure 2—figure supplement 1E-G.

– Figure 2I-L: The Authors state: "There was a decrease in the number of Runx2+ cells in the palatal mesenchyme at both E13.5 and E14.5 in Wnt1-184 Cre;Kdm6bfl/fl mice in comparison to the control" (Line 182-184). Either the Authors quantify the "number of cells" (see also comment above), or they could do a qRT-PCR to evaluate Runx2 mRNA in mutant versus control tissues.

We agree with the reviewer and have quantified both RUNX2+ and SP7+ cells in our samples. Quantification results have been added to Figure 2.

– Figure 3B: It would be beneficial to separate the genes that are upregulated from the genes that are downregulated in Wnt1-Cre;Kdm6bfl/fl mice versus controls. The overall message would be much clearer.

We thank the reviewer for this suggestion. We analyzed signaling pathways using only upregulated or downregulated genes according to this suggestion. However, after comparing the results, we think including both upregulated and downregulated genes provides better results, as both positive and negative regulators of signaling pathways are represented in the analysis.

– Figure 6: The rescue experiments are one of the main strengths of this study: beautiful and rigorous use of genetics to validate the pathways under analysis! However, it would add a great deal of strength to the rescue experiments to also evaluate the rescue at the cellular level (i.e. assess the rescue of proliferation and/or DNA damage, even if partial, or alternatively, examine whether expression of Sp7 or Runx2 is rescued by qRT-PCR).

We agree with the reviewer and appreciate this suggestion. We have assessed cell proliferation using EdU and RUNX2+ cells in our rescue model. The data has been added to Figure 7.

– Figure 7, panel I and Figure 7 – Figure Suppl 6: Red box that is described as highlighting the p53 promoter is placed instead over the promoter of a different gene, Wrap53. Indeed, this red box contains the TSS of Wrap53 and not p53. The Wrap53 gene encodes an essential component of the telomerase holoenzyme complex, a ribonucleoprotein complex required for telomere synthesis. The encoded protein interacts with other components of active telomerase and with small Cajal body RNAs (scaRNAs), which are involved in modifying splicing RNAs. It was reported that Wrap53 also functions as a p53 antisense transcript, which regulates endogenous p53 mRNA levels and the levels of P53 protein by targeting the 5' untranslated region of p53 mRNA (Mahmoudi et al. Mol Cell 2009). Therefore, Wrap53 can indirectly alter p53 levels of expression and regulation. siRNA knockdown of Wrap53 results in significant decrease in p53 mRNA and suppression of P53 induction upon DNA damage. Conversely, overexpression of Wrap53 increases p53 mRNA and protein levels (Mahmoudi et al. Molecular Cell 2009; Farnebo et al., Cell Cycle 2009; Saldana-Meyer et al., Genes and Development 2013). These studies unequivocally demonstrated that Wrap53 regulates p53. As it turns out, the primers used for the ChIP-qPCR experiment shown in Figure 7 were now blasted and they correspond to the Wrap53 promoter and not to the p53 promoter. Given all the knowledge discussed above, the current statement: "factor Tfdp1 binds to the promoter of p53 along with Kdm6b to specifically activate the expression of this tumor suppressor gene" should be re-evaluated and clarified. This point should be better investigated. As the distance between the Wrap53 and p53 genes is only approximately 1kb, in additional experiments the Authors should use primers specific to the p53 promoter and to the Wrap53 promoter together with chromatin sonicated to an average size of 300-600 bp, which should provide a clear answer as to whether TFDP1 binds directly to the p53 promoter, or to the Wrap53 promoter, or to both. If TFDP1 does not bind directly to p53 but to the Wrap53 promoter instead, regulation of p53 transcription would then be indirect, via Wrap53. This finding would still be of interest. This reviewer believes that this point is important and should be adequately clarified and examined in further depth.

We agree with the reviewer’s comment about *Wrap53*. We checked our RNAseq results, which show a very low expression level of *Wrap53*. To further confirm this result, we performed in situ RNAscope staining of our sample. The result is shown below. As our data clearly shows that *Trp53* is widely expressed in palatal mesenchyme (Figure 3), it is very unlikely that *Wrap53* is involved in the regulation of *Trp53* during palatogenesis.

– Figure 7 – Figure Suppl 6: The Authors also analyze expression patterns of Kdm6b and its co-expression with Tfdp1 by scRNAseq. The scRNAseq data sets have already been described by the same Authors in a(nother) interesting paper that was recently published. There appears to be a striking enrichment of both Kdm6b and Tfdp1 genes in a particular cell subpopulation emerging from the scRNAseq datasets. It would be very interesting to know which cell type comprises this particular cluster? Also, the individual Panels in B should be enlarged to better visualize each cell clusters. It would also be very helpful to the reader to list the specific cell types that comprise each cluster in the Panels in B.

We thank the reviewer for this suggestion. We have extracted palatal mesenchyme clusters from published data and identified these clusters with enrichment of both *Kdm6b* and *Tfdp1* expression. ScRNAseq analysis shows that cells co-expressing *Kdm6b* and *Tfdp1* are enriched in 3 clusters (*Pax9*+, *Aldh1a2*+, and *Twist1*+ cells). This result has been added to Figure 8—figure supplement 1 and the size of the images has also been adjusted.